# Temporal changes in the debris flow threshold under the effects of ground freezing and sediment storage on Mt. Fuji

Fumitoshi Imaizumi[1], Atsushi Ikeda[2], Kazuki Yamamoto[3], Okihiro Ohsaka[4]

[1]Faculty of Agriculture, Shizuoka University, Shizuoka, 422-8429, Japan
[2]Faculty of Life and Environmental Sciences, University of Tsukuba, 305-8572, Japan
[3]Bureau of Economic Affairs, Shizuoka city office, Shizuoka, 422-8429, Japan
[4]Shizuoka Professional University of Agriculture, Japan, 424-8701

*Correspondence to*: Fumitoshi Imaizumi (imaizumi@shizuoka.ac.jp)

**Abstract.** Debris flows are one of the most destructive sediment transport processes in mountainous areas because of their large volume, high velocity, and kinematic energy. Debris flow activity varies over time and is affected by changes in hydrogeomorphic processes in the initiation zone. To clarify temporal changes of debris flow activities in cold regions, the rainfall threshold for the debris flow occurrence was evaluated in Osawa failure at a high elevation on Mt. Fuji, Japan. We conducted field monitoring of the ground temperature near a debris flow initiation zone to estimate the presence or absence of seasonally frozen ground during historical rainfall events. The effects of ground freezing and the accumulation of channel deposits on the rainfall threshold for debris flow occurrence were analyzed using rainfall records and annual changes in the volume of channel deposits since 1969. Statistical analyses showed that the intensity-duration threshold during frozen periods was clearly lower than that during unfrozen periods. A comparison of maximum hourly rainfall intensity and total rainfall also showed that debris flows during frozen periods were triggered by a smaller magnitude of rainfall than during unfrozen periods. Decreases in the infiltration rate due to the formation of frozen ground likely facilitated the generation of overland flow, triggering debris flows. The results suggest that the occurrence of frozen ground and the sediment storage volume need to be monitored and estimated for better debris flow disaster mitigation in cold regions.

## 1 Introduction

Debris flows are one of the most destructive sediment transport processes in mountain areas because of their large volume and high velocity and kinematic energy (Scott et al., 2004; Theule et al., 2018; Song et al., 2019). Debris flow risks are increasing in cold mountainous regions under climate change due to changes in hydrological processes on hillslopes, increases in the magnitude of rainfall events, and those in sediment supply activity in their initiation zones (Stoffel and Huggel, 2012; Stoffel et al., 2014a, 2014b; Hirschberg et al., 2021). Because debris flows are usually triggered by heavy rainfall events (Scott et al., 2004; Sidle and Chigira, 2004; Dowling and Santi, 2014; Zhang et al., 2019), estimation of the rainfall thresholds triggering debris flows is essential for debris flow hazard mitigation measures, such as the installation of early warning systems (Pan et al., 2018; Hürlimann et al., 2019).

Many studies have attempted to determine rainfall thresholds by analyzing rainfall records in periods with and without a debris flow (e.g., Cannon et al., 2011; Hürlimann et al., 2019). The intensity-duration (ID) threshold is one of the most commonly used rainfall thresholds (Caine, 1980; Guzzetti et al., 2008; Staley et al., 2012; Zhou and Tang, 2014). Other factors, such as cumulative rainfall depth and the return time of rainfall, have also been used to estimate rainfall thresholds (Fiorillo and Wilson, 2004; Imaizumi et al., 2006). Although many studies have calculated a single rainfall threshold for each torrent and region, the threshold varies with time and is affected by changes in hydrological processes and the volume of debris flow material (Jakob et al., 2005; Schlunegger et al., 2009; Chen et al., 2012; Theule et al., 2012). Therefore, temporal changes in the rainfall threshold need to be understood to evaluate future risks of debris flows under climate change.

Debris flow activity is controlled by hydrological characteristics on hillslopes and debris flow material (i.e., channel deposits and talus slopes) in debris flow initiation zones (Bovis and Dagg, 1988; Staley et al., 2014; Loye et al., 2016). Decreases in the infiltration rate of rainfall into the ground result in the generation of overland flow that leads to debris flows (Cannon et al., 2001; Shakesby and Doerr, 2006; Ebel et al., 2012; Staley et al., 2014). The hydraulic conductivity of debris flow material also controls the initiation of debris flows (Bovis and Dagg, 1988).

In cold regions, ground freezing changes hydrological processes in watersheds (Blackburn et al., 1990; Iwata et al., 2010; Coles and McDonell, 2018). Infiltration rate and hydraulic conductivity are generally low in periods with a frozen soil layer (Blackburn et al., 1990; Iwata et al., 2010). Thus, the discharge of shallow runoff during the frozen period is higher than during the unfrozen period (Shanley and Chalmers, 1999; Coles and McDonell, 2018). Space for liquid water in soil matrix decreases by formation of ice (Zhao ang Gray, 1990; Watanabe and Flury, 2008), possibly affecting initiation condition of debris flow. However, the effect of ground freezing on the rainfall threshold of debris flow remains poorly understood.

The volume of debris flow material in the initiation zones of debris flow is controlled by the sediment supply from hillslopes and the evacuation of sediment by debris flows and fluvial processes (Imaizumi et al., 2006; Berger et al., 2011; Theule et al., 2012; Imaizumi et al., 2019; Rengers et al., 2020a). Debris flow frequency is generally high in torrents with high sediment supply activity (May, 2002; Jakob et al., 2005). Bovis and Jakob (1999) classified debris flow basins into weathering-limited basins, which require a long sediment recharge period for the occurrence of a debris flow, and transport-limited basins, in which the debris flow occurrence is primarily controlled by hydroclimatic events. These studies imply that the volume of debris flow material in the watershed needs to be interpreted to improve the accuracy of debris flow prediction.

Mount Fuji, which is the most famous mountain of Japan, has frequently experienced severe debris flow disasters, as well as slush avalanches, which are mixtures of water and snow that erode large amounts of sediment as they move downstream (Hanaoka et al., 2007). Sediment accumulated in gully bottoms, which is eroded by overland flow during rainfall events, is the main source of debris flow material. Many disasters around Mt. Fuji have occurred in spring and late autumn, despite precipitation in those seasons being lower than in summer and early autumn (Hanaoka et al., 2007). Although previous interpretations have implied that seasonal ground freezing affects temporal changes in the rainfall threshold of debris flow (Anma, 2007), this has not been supported by quantitative analyses due to the lack of field monitoring data. On Mt. Fuji, the

sediment storage volume in debris flow torrents has also changed over time because of the sediment supply from hillslopes
and the evacuation of sediment by debris flows (Hanaoka et al., 2007).

The overall aim of this study was to understand the hydrological processes triggering debris flows in relation to the formation of frozen ground and the volume of debris flow material. We did this by analyzing rainfall thresholds on Mt. Fuji. Ground freezing conditions during historical rainfall events that triggered or did not trigger debris flows and slush avalanches were estimated using an empirical model based on the field monitoring. Then, the rainfall thresholds for two periods distinguished by the presence or absence of frozen ground were firstly analysed. Secondly, influence of sediment storage volume on the rainfall thresholds was assessed both in frozen and unfrozen periods.

## 2 Study site

The Osawa failure, which is a huge gully (length 2.1 km, width 500 m, and depth 150 m), is the most active debris flow torrent on Mt. Fuji. The Osawa failure is located on the western slope of Mt. Fuji (Figs. 1a, 1b), where a westerly wind dominates. Results of age determination using [14]C indicated that this gully started to form circa 3,000 years ago (Iwatsuka and Machida, 1962). The Osawa failure increased in length and width following a catastrophic failure circa 1,000 years ago (Hanaoka et al., 2007). The total volume of sediment derived from the gully is estimated to be 75,000,000 $m^3$. The west side of Mt. Fuji is underlain by an alternation of scoria and basaltic lava, whereas thick tephra layers occur on the east side (Miyaji, 1998). Unconsolidated scoria layers have been actively eroded by freeze-thaw activity, overland flows, and strong winds. As the erosion of basal scoria layers proceeds, the lava outcrop becomes unstable due to overhanging, resulting in episodic failure of the outcrop of up to 30 m in thickness. Between 1970 to 2004, the estimated total volume of sediment evacuated from the Osawa failure was 5,020,000 $m^3$ (Hanaoka et al., 2007).

The Osawa failure has a cold climate due to its high elevation. The monthly average air temperature around the debris flow initiation zone (3,200 m a.s.l.) is lower than 0°C from November to April (Fig. 2). The maximum daily air temperature is also below 0°C from November to April. The average annual precipitation from 2010 to 2019 was 2,737 mm $yr^{-1}$. Heavy rainfall (total rainfall >100 mm) occurs during the "Baiu" rainy season (i.e., monsoon season from June to July) and during typhoon crossings (August to October). Precipitation in winter (December to February) accounts for only 8% of the annual precipitation. This relatively dry climate in winter promotes seasonal ground freezing under negative air temperatures, because thermal insulation by thick snow is inhibited except for leeward depressions, where drifted snow accumulates (Ikeda et al., 2012). Spring (March to April) is the main snowfall season on Mt. Fuji, and a study on the summit revealed that drifted snow prevented both further ground cooling and warming in the following early summer. In contrast to the leeward sites on the summit, snow insulation in the Osawa failure must be limited in the main snowfall season, because snow disappearance in mid-April, 2018, and late April, 2019, on the debris flow initiation zone (except for the narrow valley bottom) was observed by a distant-view camera installed at the western foot of the mountain by Shizuoka Prefecture.

In the Osawa failure, the slope gradients of the scoria and lava layers are 35–50° and 40–90°, respectively. Talus slopes, with a slope gradient of 35–40°, are distributed at the foot of the hillslopes. Most parts of the channel are generally covered by channel deposits, which are the main sources of debris flow material (Figs. 1c, 1d). Sediment supply processes are different between scoria and basalt. Scoria is mainly transported into channels by soil creep and surface wash, while basalt is transported by rockfall and dry ravel in relation to the failure of steep outcrops and secondary movement of deposits on hillslopes. The

channel gradient in the gully ranges from 20° (the lower part of the Osawa failure) to 35° (channel head). Debris flows and slush avalanches in the Osawa failure generally run down Osawa Creek to the Osawa alluvial fan, which is located approximately 6 km west of the gully. Some small-scale debris flows terminate in the Osawa failure and the channel sections between the gully and alluvial fan.

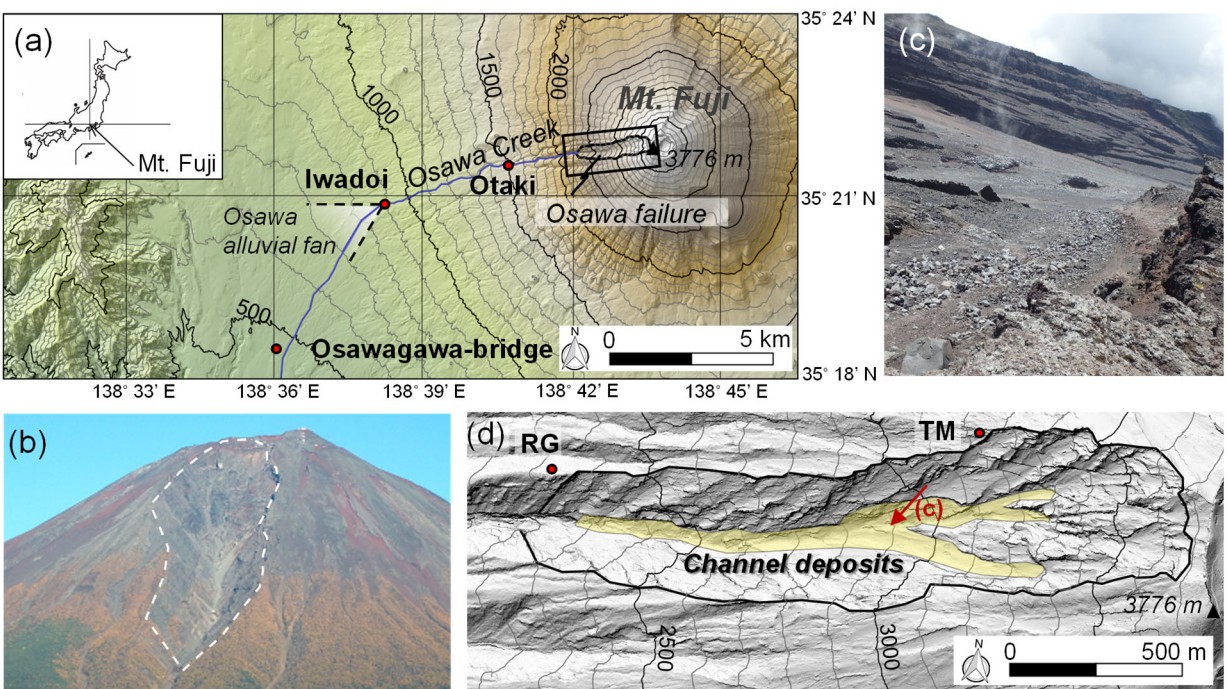

**Figure 1: Topographic map and photographs of the Osawa failure. (a) Topographic map of the area around the Osawa failure and Osawa Creek. The area shown in (d) is located within the black-outlined box. (b) Full view of Mt. Fuji. The location of the Osawa failure is indicated by the white dashed line. (c) Channel deposits in the Osawa failure. The location where this photograph was taken is indicated by the red arrow in (d). (d) Topographic map of the area around the Osawa failure. RG and TM indicate the locations of a rain gauge and ground temperature sensors, respectively. The summit meteorological station is located on the triangle**

**of the mountain peak.**

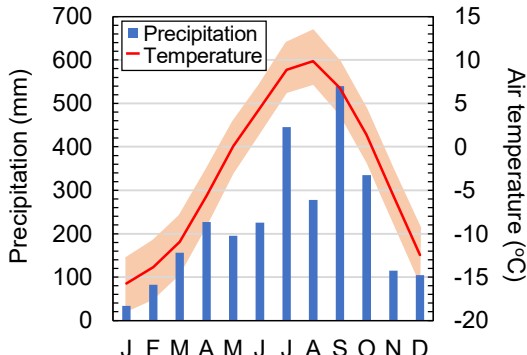

**Figure 2: Average monthly precipitation at 2,330 m a.s.l. (RG in Fig. 1d) and air temperature around the debris flow initiation zone (3,200 m a.s.l.) in the Osawa failure from 2010 to 2019. Air temperature around the debris flow initiation zone was estimated from data recorded at the summit station (3,775 m a.s.l.) and the lapse rate (0.55). The solid air temperature curve indicates monthly average temperature, while the upper and lower bounds of the shaded area indicate the average daily maximum and minimum temperatures, respectively, in each month.**

The Ministry of Land, Infrastructure, Transport, and Tourism (MLIT) has been monitoring debris flows and slush avalanches in the Osawa failure and Osawa Creek since 1969. The runout timing of previous debris flows was observed with video cameras and water level gauges at Osawagawa Bridge Observatory (490 m a.s.l.) and Iwadoi Observatory (900 m a.s.l.), as well as with wire sensors and video cameras at Otaki Observatory (1,500 m a.s.l., Fig. 1a). Some debris flows were just monitored at one or two observatories because of the absence of monitoring devices, problems encountered during monitoring (e.g., destruction of devices, electric troubles), or the short travel distance of the debris flows. The MLIT has also distinguished slush avalanches from debris flows based on field surveys just after each event and the presence or absence of snow in the flow as captured by video cameras. A total of 35 debris flows and eight slush avalanches were monitored from 1969 to 2019 (Fuji Sabo Office, 2017; Fuji Sabo Office, 2020). Four of the 35 debris flows were reclassified into slush avalanches, because snowpack over debris flow initiation zone was identified by a distant-view camera. The maximum total discharge of a single debris flow, which was estimated with a video camera, ultrasonic water level gauge, and Doppler flow-velocity sensor at Iwadoi observatory, was $3 \times 10^5$ m³ (Fuji Sabo Office, 2017).

## 3 Methodology

### 3.1 Ground temperature monitoring

A depth profile of the ground temperature was obtained at an elevation of 3,220 m a.s.l. along the northern boundary of the Osawa failure (TM in Fig. 1d). The elevation of the monitoring point was selected based on the known initiation points of many debris flows, which were determined in an analysis of aerial photographs and airborne Light Detection and Ranging (LiDAR) digital elevation models (DEMs). A 2-m-deep hole was drilled in welded scoria. Then, temperature sensors (TMC-

HD, Onset) were installed at 10 different depths (ground surface, 0.1, 0.25, 0.5, 0.75, 1.0, 1.25, 1.5, 1.75, and 2 m vertically) and connected to data loggers (U12-008, Onset). Errors in temperature measurement after calibration of the sensors were within ± 0.1°C. Monitoring was conducted from September 30, 2017 to September 17, 2019, with temperature recorded at intervals of 1 hr. The sensor at the ground surface was covered by pebbles to match its albedo to that of the natural ground surface.

The occurrence of diurnal freeze-thaw activity and the formation of seasonally frozen ground were determined based on observed ground temperature data. The period with diurnal freeze-thaw activity was determined to be the period during which the ground surface temperature rose above and fell below 0°C within 1 calendar day, whereas the period with seasonal ground freezing was determined to be the period during which the ground temperature was equal to or below 0°C over multiple days.

**3.2 Estimation of frozen periods**

To estimate the presence or absence of seasonally frozen ground during rainfall events before the monitoring of ground temperature began, the downward progress of the freezing and thawing fronts at the debris flow initiation zone since 1969 was estimated using air temperature records at the summit of Mt. Fuji (3,775 m a.s.l.), which were provided by the Japan Meteorological Agency (JMA). Because the ground-freezing period is spatially variable and affected by slope orientation, elevation, and snow distribution (e.g., Bayard et al., 2005; Kneisel et al., 2015), it is difficult to precisely estimate the frozen period at an exact debris flow initiation point, which differs among debris flow events inside the Osawa failure. Therefore, we estimated the frozen period at TM (Fig. 1d) to conduct an initial evaluation of the effect of ground freezing on debris flow occurrence.

First, the ground surface temperature at TM was estimated using a linear ordinary least squares regression for the relationship between daily average air temperature at the summit station and the daily average ground surface temperature at TM from September 30, 2017 to September 17, 2019. Although the amplitude of diurnal changes in the ground surface temperature was generally higher than that of air temperature, we used the empirical model of the relationship between air temperature and ground surface temperature due to their high correlation ($R^2 = 0.94$). Then, the downward progress of the freezing and thawing fronts was obtained using the degree-day method, which was obtained from an approximate solution of the unsteady heat conduction equation (Yamazaki, 1998; Miao et al., 2019; Walvoord et al., 2019):

$$D_f = c_f(-\textstyle\sum T(t))^{1/2}, T(t) < 0 \tag{1}$$

$$D_m = c_m(\textstyle\sum T(t))^{1/2}, T(t) > 0 \tag{2}$$

where $D_f$ is the depth of the freezing front (m), $T(t)$ is the estimated daily average ground surface temperature at TM on the $t$th day (°C d), $D_m$ is the depth of the thawing front (m), and $c_f$ and $c_m$ are coefficients derived from thermal conduction, moisture content ratio, heat of fusion and other parameters (m °C$^{-1/2}$ d$^{-1/2}$). The coefficients $c_f$ and $c_m$, which reflect temporal changes in the freezing and thawing depths as calculated from monitoring data, were used for the estimation of the freezing and thawing

fronts. This degree-day method assumes that sediment characteristics and the moisture content ratio are spatially and temporally constant.

Air temperature at the Fuji City meteorological station (20 km south of the Osawa failure), which is also maintained by the JMA, was used to estimate the ground temperature in the period for which data from the summit station were unavailable (0.9% of 1969–2019), although the elevation of this station (66 m a.s.l.) is much lower than that of the debris flow initiation zone. The regression equation for the relationship between the air temperature at the Fuji City station and ground surface temperature at the debris flow initiation zone (TM) was used to estimate the freezing and thawing lines in a depth-time graph ($R^2 = 0.93$). In this study, the periods between freezing and thawing lines are referred to as frozen periods, whereas the periods between thawing and subsequent freezing lines are referred to as unfrozen periods.

The site TM, which lacks snow cover almost all year-round, was chosen as a safe location instead of sites along the inaccessible paths of the debris flows in the Osawa failure. Except for the gully bottom, snow cover in the Osawa failure is generally shallow or absent until mid-winter, due to its slope orientation (windward side of Mt. Fuji), steep topography, and low precipitation (Fig. 2). However, snow can accumulate in spring, although no quantitative data regarding the snow distribution were available. Therefore, we preliminarily estimated the freezing and thawing lines based on the TM data. Thermal differences caused by snow cover differences between TM and the initiation points of debris flow are then discussed.

### 3.3 Rainfall monitoring

From May 1, 1969, the MLIT has observed precipitation at 2,330 m a.s.l. using a tipping-bucket rain gauge, with a precision of 1 mm, close to the Osawa failure. The rain gauge, which has an internal heating system, can also monitor winter precipitation. The observation of precipitation was occasionally interrupted by problems with the monitoring system in winter (mainly from December to April) until 2012, because of difficulties with maintenance. Hence, we could not obtain precipitation data associated with seven debris flows and one slush avalanche.

The time period used to separate different rainfall events (inter event time definition, IETD) affects the magnitude and accuracy of the rainfall threshold triggering landslides and debris flows (Bezak et al., 2016; Hong et al., 2018). The IETD in previous studies generally ranged from 6 to 24 h; it is affected by catchment size and the available data sets (Guzzetti et al., 2007, 2008; Bezak et al., 2016; Imaizumi et al., 2017). In this study, we compared rainfall thresholds obtained by three different inter event times (6, 12, and 24 h) to obtain the appropriate IETD in the Osawa failure.

Rainfall thresholds appropriately separating rainfall events with and without debris flows were determined by calculating the critical successful index (CSI, threat score) using the following equation (e.g., Liao et al., 2011; Formetta et al., 2016; Staley et al., 2017):

$$CSI = \frac{tp}{tp + fp + fn} \qquad (3)$$

where $tp$ is the number of true positive events, $fp$ is the number of false positive events, and $fn$ is the number of false negative events. In this study, the ID threshold, a widely used concept (e.g., Caine 1980; Guzzetti et al., 2008; Hürlimann et al., 2019), was employed to determine the importance of rainfall intensity and duration on the debris flow occurrence. The ID threshold is expressed by the following equation:

$$I = \alpha D^{\beta} \tag{4}$$

where $I$ is rainfall intensity, $D$ is rainfall duration, and $\alpha$ and $\beta$ are the constant and slope of the power law curve, respectively. In addition to the ID threshold producing the highest CSI value, the ID threshold for the lower boundary of debris flow occurrence was obtained by quantile regressions in the 2nd percentile (Guzzetti, 2007; 2008; Saito et al., 2010). The rainfall threshold was also obtained by comparing the maximum hourly intensity and total rainfall to further improve our understanding of the debris flow initiation mechanism. Appropriate rainfall thresholds were determined under various conditions in the debris flow initiation zone, such as the presence or absence of a frozen layer and large or small volumes of sediment storage. The uncertainty of the rainfall thresholds was accessed by 1,000 bootstrap resamplings. Total number of events in each resampling was set to the same number of rainfall events as the original observations under each condition of the debris flow initiation zone. Because we resampled events regardless of event type (i.e., debris flow, slush avalanche and no flow events), numbers of each event type were not consistent amongst resamplings. Rainfall thresholds appropriately separating rainfall events with and without sediment transfer (i.e., debris flow and slush avalanche) were obtained from each resampled dataset. Frequency distributions of the appropriate rainfall thresholds were used to evaluate uncertainty in the analysis. Debris flows and slush avalanches occur in the period with and without a snowpack in their initiation zones, respectively (Decaulne, 2007; Eckerstorfer and Christiansen, 2012). We did not distinguish between debris flows and slush avalanches to obtain rainfall thresholds, because the presence or absence of a snowpack during rainfall events was unclear, especially during events without significant sediment transfer. Rainfall events with a total rainfall <10 mm were not analyzed in this study, because they appeared to be unrelated to the occurrence of debris flows and slush avalanches. Antecedent rainfall expressed by the following equation was also calculated for each rainfall event:

$$AR = \sum_{i=1}^{n} K^{i} R_{i} \tag{5}$$

where $AR$ is antecedent rainfall, K is the decay coefficient, $R_i$ is daily precipitation in the $i$-th day preceding to the debris flow event, and $n$ is the number of days used for calculation of the antecedent rainfall. The decay coefficient K and number of days $n$ ranged from 0.7 to 1 and from 2 days to 4 weeks, respectively, in previous studies (e.g., Chen et al., 2005; Jakob et al., 2012; Guo et al., 2013; Peng et al., 2015; Bel et al., 2017). In this study, the K and $n$ were set to 0.8 and 7 days, respectively, which were commonly used values in previous studies. We adopted K smaller than 1, because a considerable amount of groundwater gradually infiltrates into deeper part of the mountain body in high elevation on Mt. Fuji due to thick clinker zones and rich cracks in basalt (Tsuchi, 2017; Tosaki and Asai, 2017).

### 3.4 Changes in the volume of channel deposits

The MLIT measured the topography of the entire Osawa failure annually using aerial photogrammetry from 1971 to 2006 and airborne LiDAR from 2007 to 2019 (Fuji Sabo Office, 2017, 2020; Table 1, Figure 3). Two measurements were conducted in each of 2000, 2004, and 2011, when large debris flows occurred. The measurements were mainly conducted from September to October, when weather conditions were stable and there was no snowpack. In the period prior to 1980, channel bed topography along cross-sectional lines with intervals of 50–100 m was assessed annually via photogrammetry using monochromatic photographs. In the period 1980–2006, DEMs with a grid size of 10 × 10 m or 5 × 10 m were constructed annually via photogrammetry using monochromatic and color aerial photographs, with scales of 1/3,000 to 1/10,000. The horizontal and vertical standard errors of the control points obtained at feature intersections in the period 1980–2006 were in the range of 0.15 to 0.30 m.

Since 2007, DEMs with a grid size of 0.5 to 1.0 m have been constructed from point clouds obtained via airborne LiDAR scanning, with a point density of 2.25 to 4.0 points m$^{-2}$. The accuracy of the measurements as determined by a comparison with ground control points on the alluvial fan was <0.1 m. The root mean square error of the elevation data between adjacent flight courses was also <0.1 m. Errors in the measurements were likely larger on steeper terrain, especially at the basalt outcrops in the Osawa failure.

Fiji Sabo Office (2020) estimated the volume of channel deposits inside the Osawa failure using a DEM in November 2000, when the channel bed elevation was lowest in large parts of the channel due to the evacuation of channel deposits by debris flows, thus determining the base topography of the channel bed (Fig. 3). In the 1974–1975, 1979–1980, and 1998–1999 periods, when topography was measured using aerial photographs, the loss of a large volume of channel deposits (>100,000 m$^3$) was estimated despite there being no debris flow signals at the Otaki, Iwadoi, and Osawagawa Bridge observatories. Errors in the estimation of sediment volume as well as the occurrence of debris flows terminating in the section between the Osawa failure and the observatories possibly affected the evaluated loss of channel deposits. Hence, our analysis of the volume of channel deposits contains some inherent errors for the period in which data were obtained using aerial photography surveys.

In this study, we obtained rainfall thresholds for two volume classes of channel deposits: ≥350,000 m$^3$ and <350,000 m$^3$. The boundary of the two classes was set to 350,000 m$^3$ to ensure that a statistically sufficient number of debris flow events could be allocated to both volume classes. Hence, this value itself does not have physical meaning relevant to the occurrence of debris flow. The inherent errors in the estimation of the volume of channel deposits may affect the misclassification of channel deposit class, especially in the period 1993–1997 when the volume of the channel deposits was close to the threshold of volume classes.

**Table 1: General description of the aerial photogrammetry and airborne LiDAR measurements**

| Period | Number of measurements | Measurement method | Data type |
|---|---|---|---|
| December 1971 to October 1979 | 9 | Aerial photogrammetry | Topography along cross-sectional lines with intervals of 50–100 m |
| October 1980 to November 2000 | 22 | Aerial photogrammetry | DEM with a grid size of 10 × 10 m |
| September 2001 to October 2006 | 7 | Aerial photogrammetry | DEM with a grid size of 10 × 5 m |
| October 2007 to October 2019 | 12 | Airborne LiDAR | DEM with a grid size of 1.0 m |

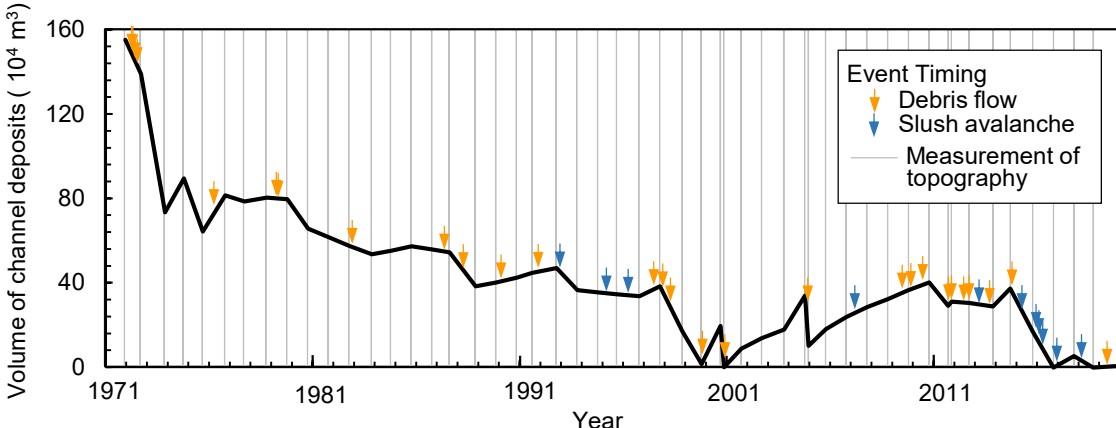

**Figure 3: Temporal changes in the volume of channel deposits in the Osawa failure (original data from Fuji Sabo Office, 2020).**

## 4 Results

### 4.1 Changes in ground temperature

In the shallower part of the ground (i.e., ≤0.25 m deep), diurnal freeze-thaw activity was frequently observed in autumn and spring, when the diurnal amplitude of the ground surface temperature was evident, whereas seasonally frozen ground was observed in winter (Figs. 4 and 5). At depths between 0.25 to 1.25 m, seasonally frozen ground occurred over long periods up to four months (Fig. 5). The minimum ground temperatures recorded each winter were lowest at the surface and increased toward the deeper part of the ground (Fig. 4), and the ground below 1.75 m was not frozen throughout the monitoring period (Fig. 5). Temporal patterns in the ground temperature were different between the 2017–2018 and 2018–2019 frozen periods (Fig. 4). In 2017–2018, ground temperature above a depth of 1.5 m rapidly increased during a heavy rainfall event on March 5 (total rainfall and maximum hourly rainfall of 89 mm and 18 mm h$^{-1}$, respectively), which triggered a slush avalanche.

Although the seasonally frozen layer was retained for more than 1 month after the rainfall event, the ground temperature at a depth >0.5 m did not fall much below 0°C. In spring 2019, when heavy rainfall events were not observed, the ground temperature below 0.5 m was lower than that in spring 2018. After ground warming on March 20–21 and April 20–24, 2019, ground cooling was observed, because the spring was colder than in the previous year. In both winters, seasonally frozen ground to a depth of 1.25 m persisted until mid-June in 2018 and early July in 2019 (Fig. 4). The clear diurnal changes in ground surface temperature in both the 2017–2018 and 2018–2019 frozen periods indicated that the snowpack was never sufficiently deep to prevent ground cooling in the observed winters.

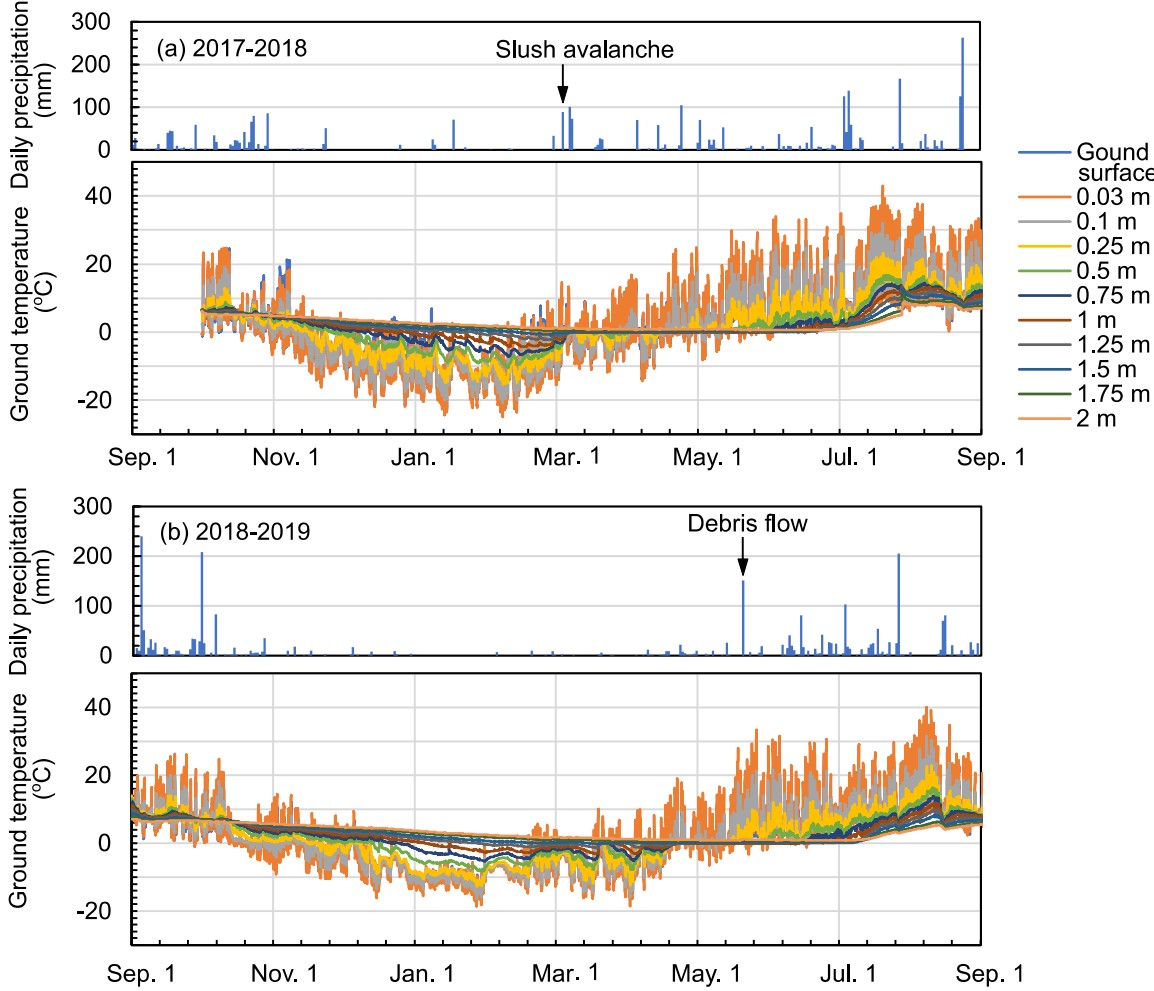

**Figure 4: Temporal changes in the ground temperature at 3,220 m a.s.l. from September 1, 2017 to September 1, 2019. Daily precipitation is also shown in the figure.**

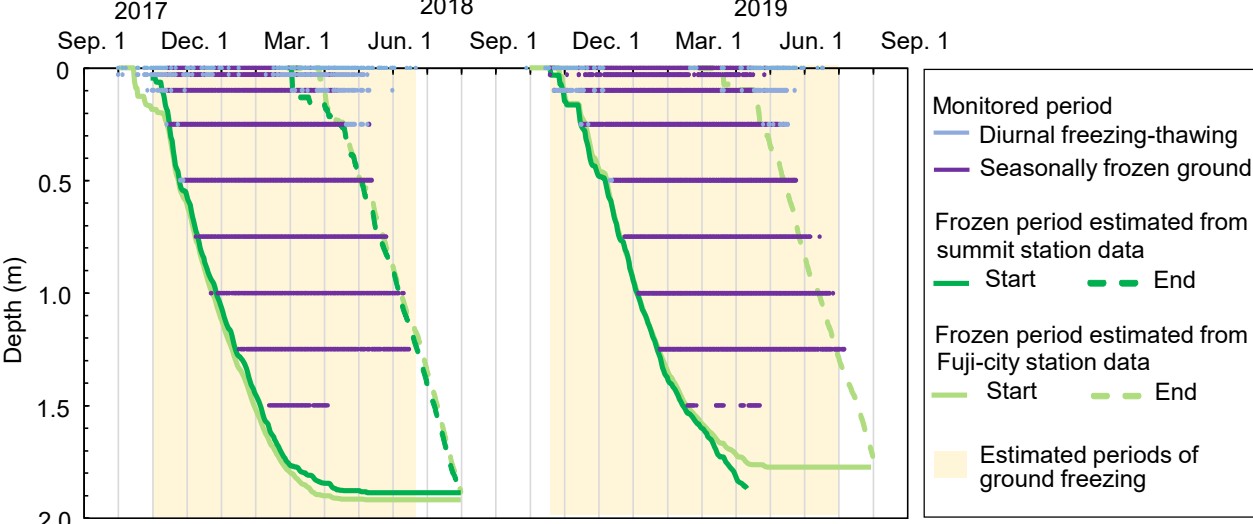

**Figure 5: Comparison of observed and estimated periods of diurnal freeze-thaw activity and seasonal ground freezing at 3,221 a.s.l. The timing of thawing in 2019 could not be estimated using data from the summit station due to monitoring equipment failure. Periods of ground freezing were estimated using summit station data, whereas the end of the frozen period in 2019 was estimated using data from the Fuji City station.**

The most appropriate coefficient $c_f$ that gave the minimum standard error between the monitored and estimated timing of freezing in the depths between 0.25 to 1.25 m (= 3.6 days) was 0.055 m °C$^{-1/2}$ d$^{-1/2}$ (Fig. 5). Except in the near-surface (<0.25 m) and deep layers (>1.5 m), the positions of the freezing front were adequately estimated based on air temperatures measured at the summit and Fuji City stations (Fig. 5). The disagreement between the estimated freezing and observed non-freezing states at the depth of 1.75 m implied the presence of a thermal buffer, such as a layer with a high water content.

The rapid and simultaneous increase in ground temperature from negative values to the melting point (Fig. 4) indicated that thawing of the porous volcanic ground was controlled by not only heat conduction but also heat advection. In particular, infiltrated rainwater on March 5, 2018 passed through the seasonal frozen layer and released latent heat by partly freezing in the voids (Fig. 4). Ikeda et al. (2012) observed partial but rapid thawing of the frozen layer on the summit due to heavy rainfalls, mostly from tropical cyclones, in late summer, although thawing due to rains in early summer was unclear at TM in 2018 and 2019 (Fig. 4). Taking this into account, we calculated the coefficient $c_m$ as an empirical value rather than a physical parameter. The empirical $c_m$ was 0.068 m °C$^{-1/2}$ d$^{-1/2}$ when Eq. (2) best reproduced the time of thawing at a depth of 1.25 m (Fig. 5).

The 1.5 m-deep point thawed much earlier than the 1.25 m-deep point (Fig. 5), which indicates that geothermal heat thawed ground upward around the deepest position of the seasonal frozen layer. However, most of the frozen layer thawed downward because the heat input from the surface was much larger than the geothermal heat flow. In this study, periods of ground freezing could be roughly estimated from the time when the freezing front started to progress downward from the ground surface, and the time when the estimated thawing front reached a depth of 1.25 m (Fig. 5).

## 4.2 Seasonal pattern of debris flow occurrence

In the Osawa failure, the rainfall pattern is different between frozen and unfrozen periods (Fig. 6). In the frozen periods, hourly rainfall intensity exceeds 30 mm h$^{-1}$ several times each year, while the hourly rainfall intensity seldom exceeds 30 mm h$^{-1}$ in frozen periods (Fig. 6). Antecedent rainfall in unfrozen periods is also higher than in frozen periods. Antecedent rainfall sometimes exceeds 150 mm in unfrozen periods, while the antecedent rainfall is generally lower than 100 mm in frozen periods (Fig. 6).

The numbers of debris flows and slush avalanches were highest in November and March, respectively (Fig. 7). Although average monthly precipitation in August (278 mm) was higher than in November (115 mm; Fig. 2), no debris flow was observed in August. Only one debris flow and one slush avalanche occurred in mid-winter (i.e., January and February), when precipitation was mostly snow. Debris flows occurred during rainfall events with high hourly rainfall intensity (Fig. 6). However, it is difficult to clearly separate debris flows and no flow events based on a single rainfall index. For example, on September 4, 2011, debris flow was not triggered by a rainfall event with high rainfall intensity (38 mm h$^{-1}$) and high total rainfall (595 mm) as well as 62 mm of antecedent rainfall prior to the event. In contrast, on November 19, 2011, a debris flow was triggered by low rainfall intensity (19 mm h$^{-1}$), moderate total rainfall (164 mm), and no antecedent rainfall.

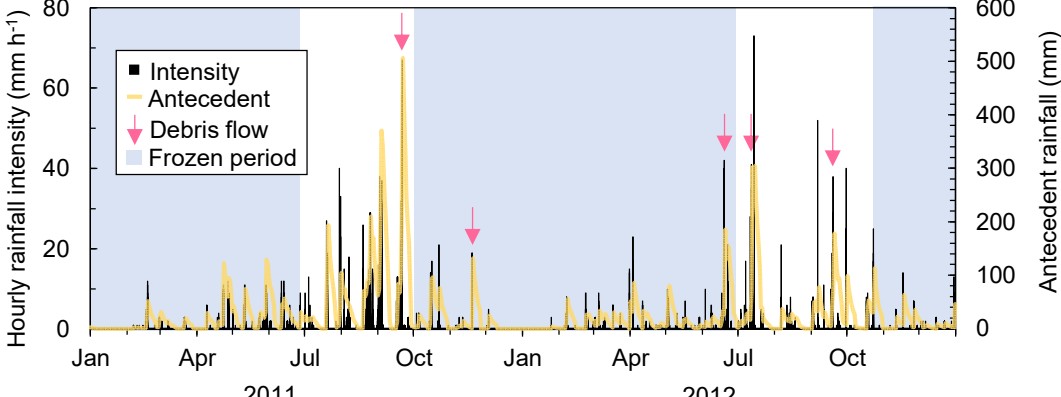

**Figure 6: Seasonal changes in the rainfall pattern in the period 2011–2012, when debris flows occurred successively. The timing of the debris flow events is also illustrated. Frozen periods were estimated by the degree-day method. Antecedent rainfall was determined using Eq. (5).**

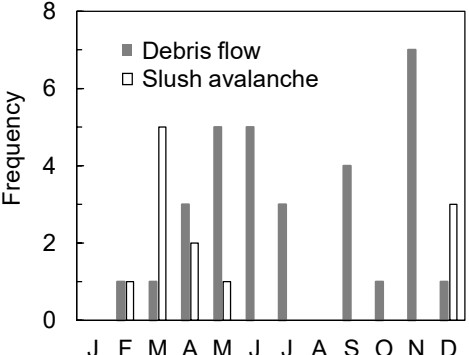

**Figure 7: Monthly frequency of debris flows and slush avalanches.**

### 4.3 Effects of ground freezing on the conditions required to initiate a debris flow

Figures 8a to 8c show the influence of rainfall intensity and duration on sediment transfer (i.e., debris flows and slush avalanches), which tended to occur during the conditions represented in the upper-right quadrant of all plots regardless of the IETD. This indicated that the magnitude of rainfall events that triggered sediment transfer was larger than that of an ordinary rainfall event. However, the distributions of rainfall events that did or did not trigger sediment transfer could not be completely separated. The length of the IETD affected the rainfall duration rather than the rainfall intensity, with rainfall duration

becoming longer with an increasing IETD. The CSI for the ID thresholds was significantly low (<0.25). The slope of the power law curve was gentler with increasing IETD. Among the three IETDs, 24 h resulted in the highest CSI value. Much of the sediment transfer occurred during the event with low antecedent rainfall (i.e., <50 mm). On the other hand, sediment transfer did not occur during some rainfall events with high antecedent rainfall (i.e., ≥100 mm).

     Rainfall thresholds for the occurrence of sediment transfer were also obtained by comparing total rainfall and maximum

hourly rainfall intensity (Figs. 8d–8f). Regardless of the IETD, the total rainfall and maximum rainfall intensity that produced the highest CSI value were 206 mm and 42 mm h$^{-1}$, respectively. The distributions of rainfall events that did or did not trigger sediment transfer could not be separated well (CSI < 0.17). The distribution of no flow events with high antecedent rainfall (i.e., ≥100 mm) overlapped with that of sediment transfer events with low antecedent rainfall (i.e., <50 mm). Although the amount of antecedent rainfall is affected by the setting of K and $n$ in Eq. (4), Fig. 8 implies that the antecedent rainfall was not

a critical factor affecting the occurrence of sediment transfer. The CSI did not differ among the IETDs (Figs. 8d–8f). Hereafter, the IETD was set to 24 h, which resulted in the highest CSI in the intensity-duration analysis.

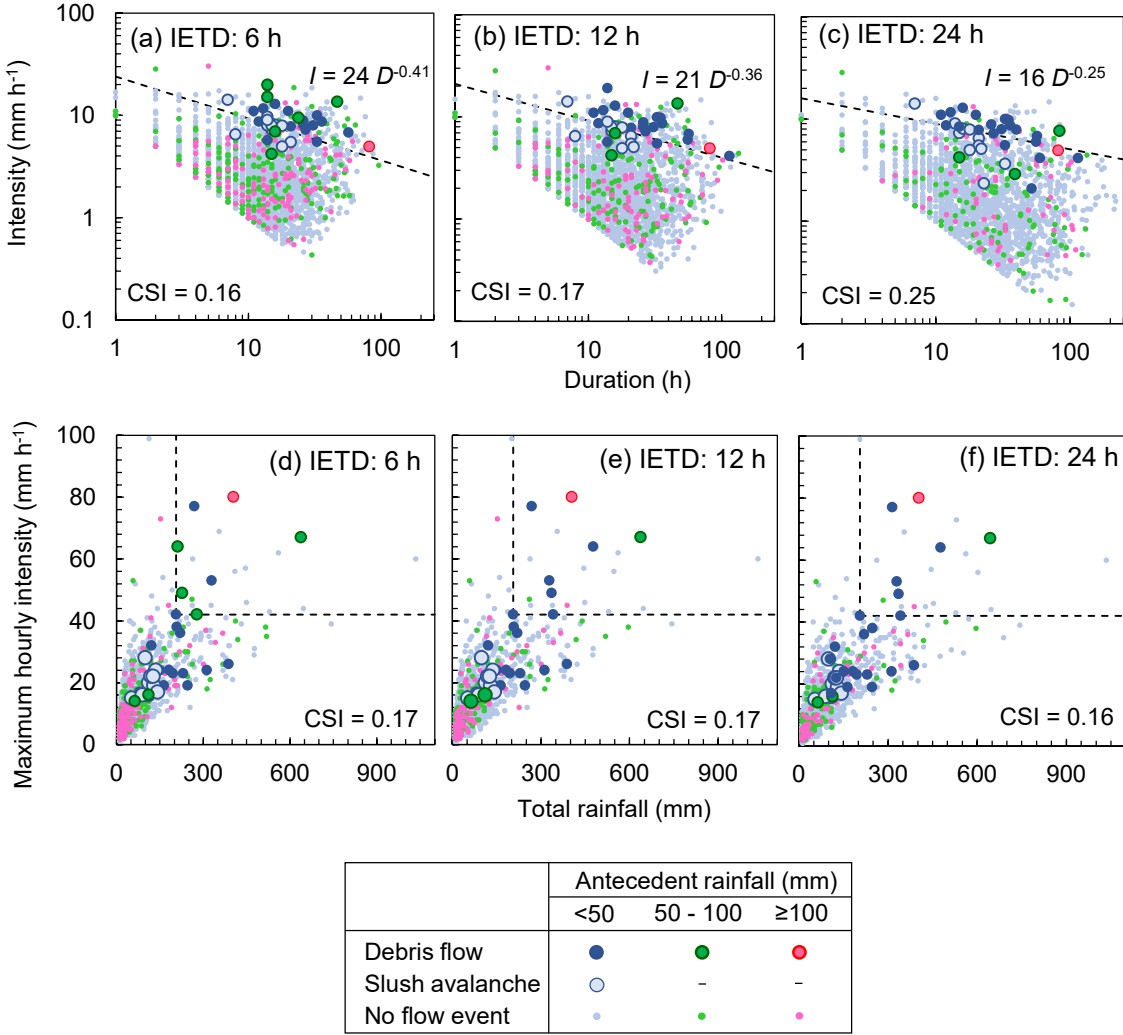

**Figure 8: Rainfall thresholds for triggering debris flows and slush avalanches. Comparison of the duration and intensity of rainfall events with or without debris flows and slush avalanches that were calculated with an inter event time definition (IETD) of (a) 6 h, (b) 12 h, and (c) 24 h. Dashed lines indicate the rainfall thresholds with the highest critical successful index (CSI). Comparison of the total rainfall and maximum hourly rainfall intensity of rainfall events with or without debris flows and slush avalanches that were calculated with an IETD of (d) 6 h, (e) 12 h, and (f) 24 h.**

When separate ID diagrams were plotted for the frozen and unfrozen periods, the distribution of rainfall events that triggered sediment transfer during frozen periods was positioned further to the lower-left side than in the corresponding diagram for unfrozen periods (Figs. 9a). Thus, the ID threshold in frozen periods was lower than that in unfrozen periods. The CSI values of the ID thresholds in the frozen and unfrozen periods were 0.42 and 0.16, respectively. Therefore, the CSI in frozen periods was substantially improved compared to the CSI for all periods (0.25, Fig. 8c). The frequency distribution of constant α in Eq.

4 was obtained by 1,000 bootstrap resamplings and clearly differed between the frozen and unfrozen periods, while slope β widely overlapped between the two periods (Figs. 9b, 9c). The lower limit of rainfall triggering sediment transfer, which was obtained by the 2nd percentile regression lines, in the frozen periods was lower than that in the unfrozen periods over a wide range of rainfall durations. This also indicated that the rainfall threshold for sediment transfer in the frozen periods was lower than that in the unfrozen periods. Some no flow events were distributed in the upper-right side of the sediment transfer plots in the ID diagram, especially in the unfrozen periods. Therefore, the upper limit of rainfall events did not produce sediment transfer could not be determined in the Osawa failure. The distribution of slush avalanches, which were observed only during frozen periods, overlapped with that of debris flows.

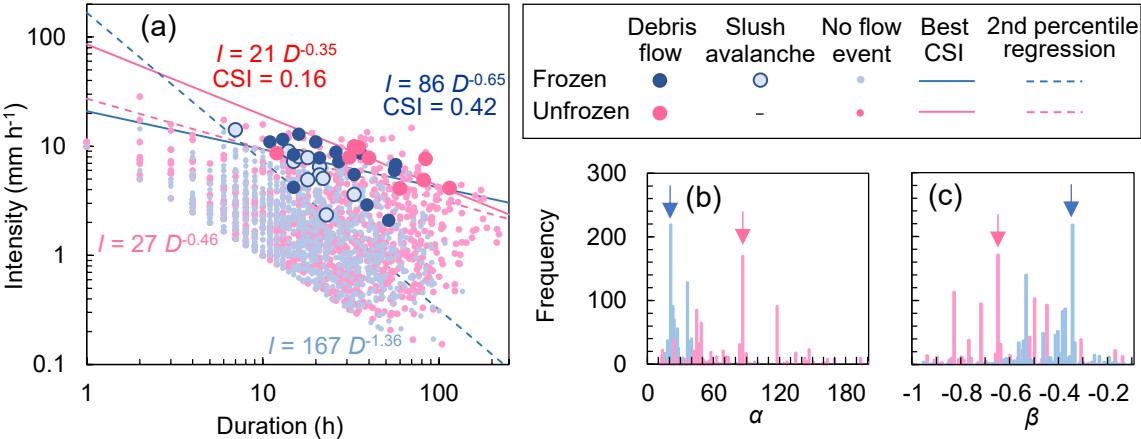

**Figure 9: Intensity-duration thresholds for triggering debris flows and slush avalanches. (a) Comparison of the duration and intensity of rainfall events with or without debris flows and slush avalanches for rainfall events in frozen and unfrozen periods. Solid lines in the plots indicate the rainfall threshold, which optimally separated rainfall events that induced debris flows and slush avalanches from those that did not. Dashed lines in the plots indicate 2nd percentile regression lines that represent the lower limit condition for the occurrence of sediment transfer. Frequencies of (b) constant α and (c) the slope of the power law curve β, which produced the highest critical successful index (CSI), obtained by 1,000 bootstrap resamplings are also shown in the figure. Arrows indicate values that produced the highest CSI in the analysis using the original data set (solid line in (a)).**

As with the ID diagrams, the maximum hourly intensity and total rainfall of rainfall events in the frozen and unfrozen periods were plotted separately (Figs. 10a). In the frozen periods, the threshold values of total rainfall and maximum hourly rainfall intensity were 99 mm and 19 mm h$^{-1}$, respectively. The slush avalanches were distributed in the lower-left side of the diagram compared to the debris flows. In the unfrozen periods, the threshold values of total rainfall and maximum hourly rainfall were higher than in the frozen periods (314 mm and 64 mm h$^{-1}$, respectively). Both of the debris flows triggered by a maximum hourly rainfall <30 mm in unfrozen periods occurred within the 10-day period prior to the estimated subsequent frozen period. Thus, it is possible that the ground was frozen when these debris flows occurred. The CSI values of the rainfall thresholds in the frozen and unfrozen periods (0.38 and 0.36, respectively) were better than those obtained using all rainfall

events (0.16; Fig. 8f). The frequency distributions of maximum hourly intensity and total rainfall obtained by bootstrapping
did not overlap between the frozen and unfrozen periods (Figs. 10b, 10c), indicating that the difference in threshold was not
affected by the presence or absence of specific rainfall events.

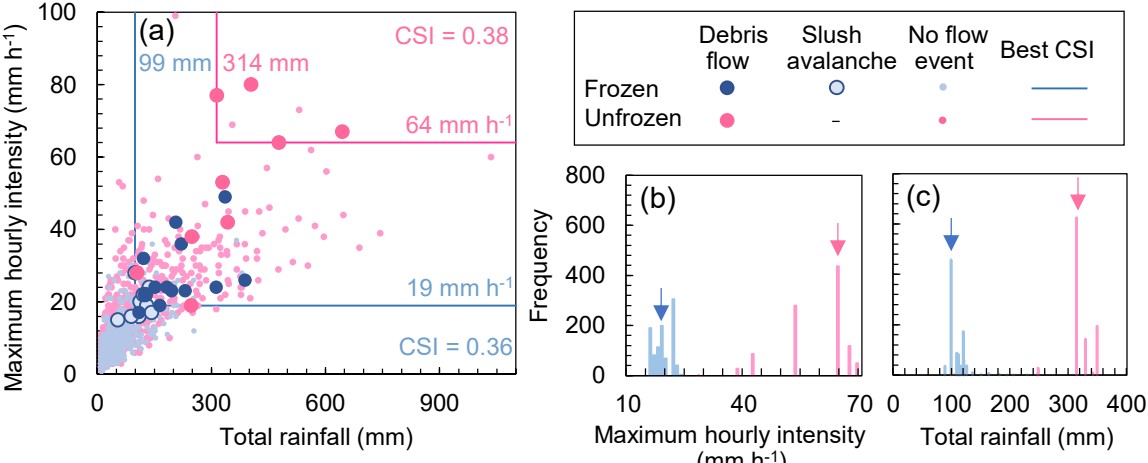

**Figure 10: Thresholds of total rainfall and maximum hourly rainfall intensity triggering debris flows and slush avalanches.** (a)
Comparison of the total rainfall and the maximum hourly rainfall intensity of rainfall events with or without debris flows and slush
avalanches for rainfall events in the frozen and unfrozen periods. The solid lines in the plots indicate the rainfall thresholds that
optimally separated rainfall events that induced debris flows and slush avalanches and those that did not. Frequencies of (b)
maximum rainfall intensity and (c) total rainfall that produced the highest critical successful index (CSI), obtained by 1,000
bootstrap resamplings. Arrows indicate the values that produced the highest CSI in the analysis using the original data set (same as
values in (a)).

**4.4 Effects of the volume of channel deposits on the conditions required to initiate a debris flow**

During frozen periods, the ID threshold in the periods with large volume of channel deposits was higher than in periods with
small volumes of channel deposits (Fig. 11a). However, as a result of bootstrapping, the frequency distributions of both α and
β overlapped between periods with large and small volumes of channel deposits (Figs. 11b, 11c). Hence, a high uncertainty
exists in the effect of the volume of channel deposits on the ID threshold. The CSI values of the thresholds for small and large
volumes of channel deposits were 0.42 and 0.46, respectively. Therefore, the CSI was not improved from when the ID threshold
was determined without considering the volume of channel deposits (0.42, Fig. 9a). During unfrozen periods, the ID threshold
in periods with a large volume of channel deposits was slightly higher than in periods with a small volume of channel deposits
(Fig. 11d). As with the situation during frozen periods, the frequency distributions of both α and β were overlapped between
periods with large and small volumes of channel deposits (Figs. 11e, 11f). The CSI values for both volume groups (0.20 and
0.14 for small and large volumes of channel deposits, respectively) were not significantly higher than the values obtained
without considering the volume of channel deposits (0.16, Fig. 9a).

As with the ID threshold, the maximum hourly intensity and total rainfall that triggered debris flows were not clearly affected by the volume of channel deposits during the frozen periods (Figs. 12a). Frequency distributions of maximum hourly rainfall and total rainfall in bootstrapping had a clearer peak than those of α and β of the ID threshold (Figs. 12b, 12c). The thresholds of maximum hourly rainfall intensity in the unfrozen periods were different and were dependent on the volume of channel deposits (Figs. 12d, 12e). A larger hourly rainfall intensity was needed for the occurrence of debris flows in periods with a large volume of channel deposits. Many rainfall events that exceeded the hourly intensity threshold in periods with a small volume of deposits did not trigger debris flows in periods with a large volume of deposits. The threshold of total rainfall during unfrozen periods was less clear than that of hourly rainfall intensity (Figs. 12d, 12f).

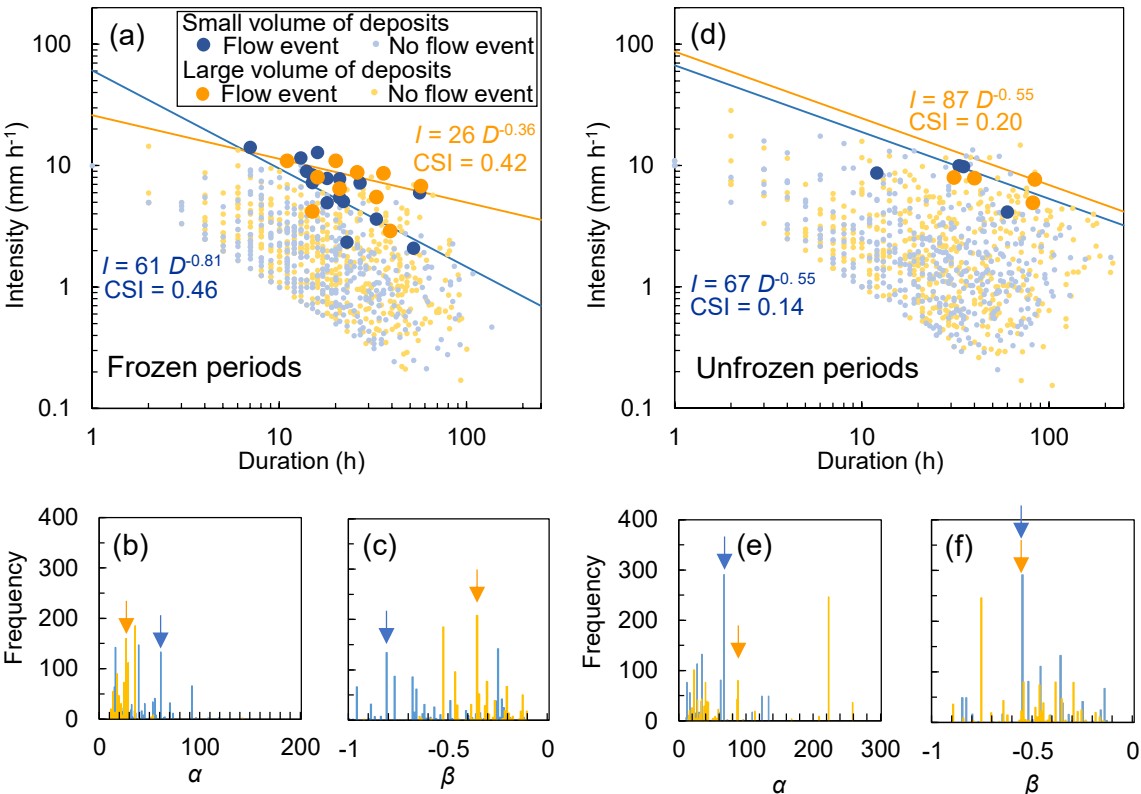

**Figure 11: Comparison of the duration and intensity of rainfall events with or without debris flows and slush avalanches, analyzed according to the volume of channel deposits. The boundary between the two volume classes was set to 350,000 m³. (a) Comparison during frozen periods. The frequencies of (b) constant α and (c) slope of the power law curve β, which produced the highest critical successful index (CSI), obtained by 1,000 bootstrap resamplings during frozen periods. (d) Comparison during unfrozen periods. (a) Comparison during frozen periods. Frequencies of the (e) constant α and (f) slope of the power law curve β obtained by bootstrap resampling during unfrozen periods.**

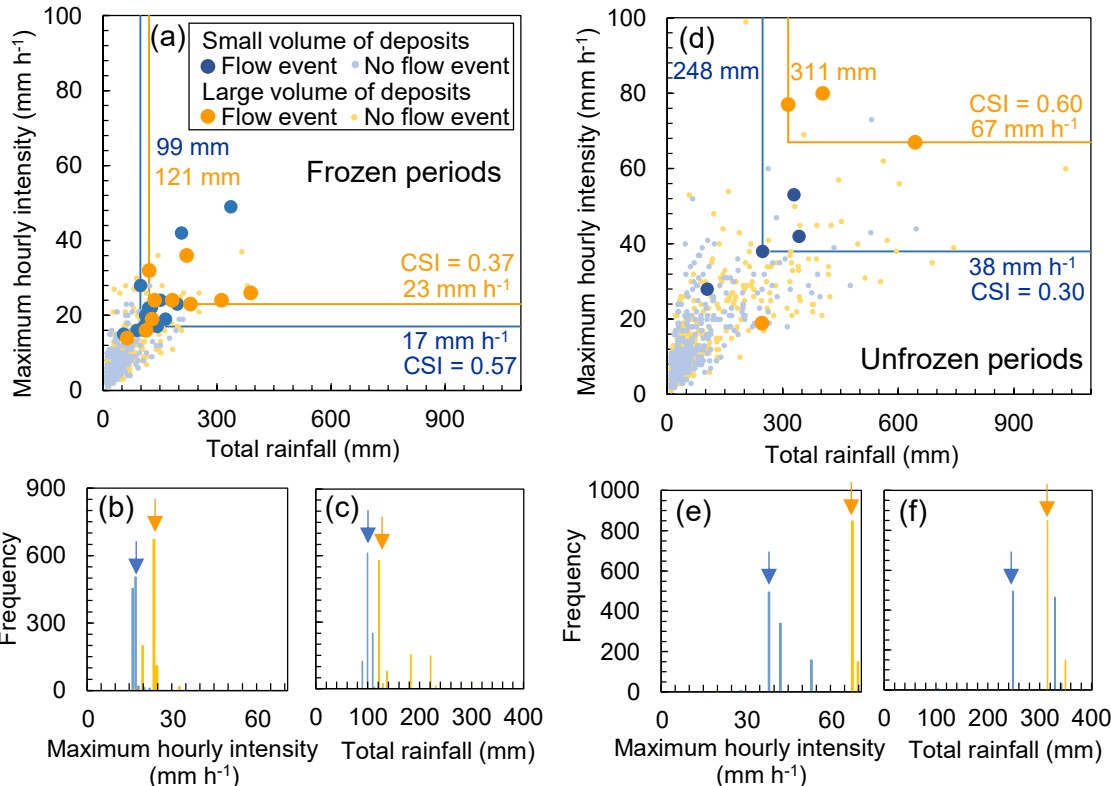

**Figure 12: Comparison of the total rainfall and maximum hourly rainfall in rainfall events with or without debris flows and slush avalanches, plotted according to the volume class of channel deposits. The boundary between the two volume classes was set to 350,000 m³. (a) Comparison during frozen periods. The lines in the plots indicate the rainfall thresholds that optimally separated rainfall events that induced debris flows and slush avalanches and those that did not. Frequencies of (b) maximum hourly rainfall intensity and (c) total rainfall, which produced the highest critical successful index (CSI), obtained by 1,000 bootstrap resamplings during frozen periods. (d) Comparison during unfrozen periods. Frequencies of (e) maximum rainfall intensity and (f) total rainfall obtained by bootstrap resampling.**

## 5. Discussion

Our simple model using ground surface temperature could roughly reproduce the period of ground freezing at site TM (Fig. 5), although subsurface heterogeneity, infiltration of rainfall, and upward movement of sensible heat were not considered in the model. The snowpack, which affects the period of ground freezing in many cold regions (e.g., Zhang, 2005; Luo, 2018), was not considered important to the formation of frozen ground at the monitoring point, because diurnal changes in the ground surface temperature were clear even in mid-winter (Fig. 4). Similarly, strong winds under the dry climate in winter probably prevented snow accumulation on the windward steep slopes such as the whole Osawa failure (Ikeda et al., 2012), which likely resulted in the limited influence of the snowpack on ground freezing observed in this study. However, snowfalls from March

to April may affect the thermal situation differently among topographic conditions. Air temperatures from March to April fluctuated around 0 °C, which had a direct impact on the diurnal freeze-thaw of the near-surface at site TM (Fig. 4). This situation hindered further downward freezing at the bottom of the frozen layer and thawing below the diurnal freeze-thaw layer (Fig. 5). In the same period, where there was thick snow insulating the ground from the atmosphere, the snow must have prevented both further cooling and thawing of the existing frozen ground. Thus, the thickness of the frozen layer formed in winter might be independent of the snow thickness in spring. In addition, the ground at TM thawed after snow disappearance in the debris flow initiation zone (mid-April, 2018 and late April, 2019). These conditions suited our expansion of the simplistic thermal model to distinguish between frozen and unfrozen periods over the initiation zone. Although the ground freezing and snow conditions of channel deposits at very bottom of the valley may differ from those at TM, the effect of ground freezing could be demonstrated by the estimation of freezing periods at TM (Figs. 9, 10). This implies that the water supply from hillslopes, which have a greater area than the valley bottom, is important to the occurrence of sediment transfer.

Because hourly rainfall intensity and total rainfall in the unfrozen period was higher than in frozen periods (Figs. 6, 9), sediment transfers can occur more easily in the frozen periods in terms of the rainfall characteristics. However, the amount of sediment transfer in frozen periods was larger than that in unfrozen periods (33 and 9, respectively) because of the lower rainfall thresholds in the frozen periods (Figs 9, 10). Therefore, the formation of frozen ground is a critical factor affecting seasonal changes in the susceptibility to sediment transfer in the Osawa failure. In unfrozen periods, many rainfall events exceeding rainfall thresholds in frozen periods did not trigger sediment transfer, also implying the importance of ground freezing in the occurrence of sediment transfer (Figs 9, 10). Bovis and Dagg (1988) implied that the hydraulic conductivity of channel deposits affects the temporal changes in debris flow activity. Based on constant head permeability tests in the laboratory, Nanzaka and Iwata (1989) reported that the hydraulic conductivity of frozen deposits in the Osawa failure (3 to 22 mm h$^{-1}$) was lower than that of unfrozen deposits (720 mm h$^{-1}$). Decreases in the infiltration rate due to ground freezing, which results in the occurrence of overland flow, have been reported in other regions (Blackburn et al., 1990; Iwata et al., 2010; Coles and McDonell, 2018). Consequently, the generation of overland flow that was affected by the low infiltration rate of deposits in frozen periods likely resulted in debris flow occurrence during smaller rainfall events (Fig. 10). Ice melt, which increases the water supply to debris flow material, caused by rainfall is another potential factor lowering the rainfall threshold. The effect of the volume of channel deposits on the rainfall threshold was unclear in the frozen periods (Figs. 11a, 12a), suggesting that debris flows in these periods were triggered by hydrological processes near the ground surface rather than the saturation of entire channel deposits (Fig. 13a). Increasing debris flow frequency due to decreases in the infiltration rate has also been reported in regions affected by wild fire (Staley et al., 2017; McGuire et al., 2018; Rengers et al., 2020b). Because a rainfall index representing the total water supply to the basin (i.e., total rainfall) was also an important factor in the triggering of debris flows (Fig. 10a), a sufficient stream discharge may also be needed for the occurrence of debris flow.

Slush avalanches and debris flows have been recorded in cold regions other thatn the Osawa failure (Decaulne and Saemundsson, 2006; Decaulne, 2007). In contrast to a debris flow, the presence of a snowpack is necessary for the occurrence of a slush avalanche (Decaulne, 2007; Eckerstorfer and Christiansen, 2012). The magnitude of rainfall events triggering slush

avalanches were smaller than those that triggered debris flows (Fig. 10a), because snowmelt during rainfall events increases the water supply to debris flow material (Decaulne and Saemundsson, 2006). Because snow in a slush avalanche sometimes melts as it moves to lower elevations due to the increasing temperature (Anma, 2007), some sediment transfer events classified as debris flows would originally have been slush avalanches in the initiation zone. That could be one of the reasons why the distributions of slush avalanches and debris flows according to rainfall factors overlapped in Fig. 9a.

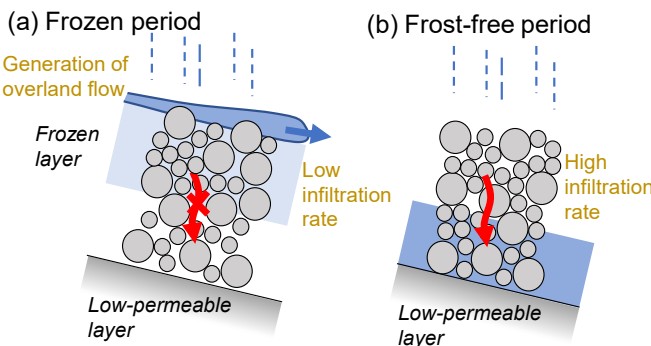

**Figure 13: Schematic diagram of the hydrological characteristics of the Osawa failure. (a) Frozen periods. The layer with the ice in soil matrix is colored by light blue. (b) Unfrozen periods.**

In contrast to the frozen periods, overland flow rarely occurred in the unfrozen periods because of the high infiltration rate of the deposits. Therefore, an increase in the groundwater level in channel deposits would likely influence the occurrence of debris flows (Fig. 13b). The slope of the ID threshold in environments in which the rainfall duration was important was steeper than in environments in which rainfall intensity was important (Guzzetti et al., 2008). The steeper threshold line in the unfrozen periods relative to the frozen periods (Fig. 11) indicated that rainfall duration was the important factor in the unfrozen periods. A high hourly rainfall intensity was needed for the occurrence of debris flows when a large volume of sediment was stored in the initiation zone (Fig. 12b). The relationship between the rainfall threshold and the volume of sediment stored was different from that reported in weathering-limited debris flow basins (Bovis and Jakob, 1999; Larsen et al., 2006). Because debris flows occurred even in periods with a small volume of deposits stored in the channel (Fig. 3), the threshold of sediment volume needed for the occurrence of a debris flow, which has been reported in weathering-limited basins (Bovis and Jakob, 1999), was not clear in the Osawa failure. The relationship between the rainfall threshold and the storage volume in the Osawa failure was also different from that in transport-limited basins, in which the rainfall threshold can be given as a constant value regardless of the storage volume (Bovis and Jakob, 1999; Schlunegger et al., 2009). Channel deposits in the Osawa failure had a thickness of up to 10 m in some periods and were also characterized by a high infiltration rate. The ratio of the thickness of the saturated layer to the entire thickness of the deposits is a factor controlling the movement of cohesionless deposits in steep channels, because the ratio determines the balance between shear stress and shear strength (Imaizumi et al., 2016, 2017). A

large water supply is needed for the movement of thick deposits. Hence, the volume of channel deposits likely has a negative effect on the debris flow occurrence in unfrozen periods. In case of exceedance of the rainfall thresholds, however, it is possible that the debris flow volumes become larger when a large volume of channel deposits is stored in the channel. In addition, this specific characteristic of the rainfall threshold in unfrozen periods could also result from the fact that there are relatively few triggering events.

Plots of rainfall events with and without debris flow events overlapped to a certain extent, even after consideration of the frozen periods and volume of channel deposits (Figs. 11, 12). The magnitude of some rainfall events that triggered debris flows in unfrozen periods was as low as those in the frozen periods (Fig. 10), indicating that it is was difficult to accurately estimate when frozen periods occur. Because periods of ground freezing differ with elevation, it is difficult to determine if the ground was frozen at the initiation point for all events. Another potential factor reducing the CSI was the spatial variability of precipitation in the Osawa failure, which has a difference of elevation of up to 1,500 m. Although most of the channel deposits in the Osawa failure consistently face west (Fig. 1d), slope aspect controls debris flow activity in other regions (Tillery and Rengers, 2020; Ding et al., 2020) due to its impact on the duration of seasonal ground freezing (Dashtseren et al., 2014), weathering rate of the rock wall (Rengers et al., 2020a), distribution of water (Xu et al., 2013), and geometric characteristics (Ding et al., 2020). The ID threshold for the Osawa failure was higher than those measured for many other torrents globally (e.g., Guzzetti et al., 2008; Bezak et al., 2016; Hürlimann et al., 2019). Although the lower limit for debris flow in the Osawa failure was not far from the ID thresholds for other regions, rainfall levels above the lower limit without debris flow occurrence pushed the threshold upward. The observatories are located far from the debris flow initiation zone, resulting in debris flows just a short distance away being missed. This might have also affected the higher threshold line.

Under climate change, increases in the amount of sediment delivered to channels by the thawing of permafrost and increases in extreme rainfall events are considered to be potential factors increasing the threat of debris flows on cold mountains (Staffler et al., 2008; Stoffel et al., 2014a, 2014b). Climate change also provide sediment in torrents where there was no loose sediment before. On the other hand, results of our study imply that climate change also impacts seasonal debris flow activity via shifts in the periods of seasonal ground freezing. Because rainfall thresholds in unfrozen periods were higher than that in frozen periods (Figs. 9, 10), global warming decreases debris flow risks by shortening the period of seasonal ground freezing. Increased amount of sediment delivered to channels by thawing of permafrost possibly increases volume of sediment storage in debris flow initiation zone (Stoffel et al., 2014b), resulting in the higher rainfall threshold for the occurrence of debris flow in unfrozen periods (Fig. 12b). Recently, a stochastic weather generator model by Hirschberg et al. (2021) predicted future decrease in frequency and sediment yield of debris flow as a result of decrease in sediment supply from hillslopes. Consequently, future climate change has both positive and negative effects on debris flow hazards.

## 6. Summary and Conclusion

To determine the effects of frozen ground on temporal changes in the rainfall threshold for the occurrence of debris flows and slush avalanches, we estimated periods of seasonal ground freezing based on field monitoring conducted close to the initiation zone in the Osawa failure on Mt. Fuji. We also evaluated the influence of sediment storage volume in the debris flow initiation zone on the rainfall threshold. The ID threshold in periods of ground freezing was clearly lower than in periods without ground freezing. A comparison of the maximum hourly rainfall intensity and total rainfall during rainfall events also showed that debris flows occurred during periods of seasonal ground freezing with rainfall events of a smaller magnitude. Decreases in the infiltration rate due to the formation of frozen ground facilitated the generation of overland flow, which triggered debris flows. Global warming possibly decreases debris flow risks by shortening the period with seasonal frozen ground, which decreases rainfall threshold for the occurrence of debris flow.

The volume of channel deposits also affected the rainfall threshold, especially during unfrozen periods. The threshold of hourly rainfall intensity in periods with large volumes of channel deposits was higher than that in periods with small volumes of channel deposits, indicating that a large water supply is needed for the debris flow occurrence when channel deposits are thick. Increases in the water content in channel deposits caused by the infiltration of rainfall are important for the debris flow occurrence during unfrozen periods, whereas hydrological processes near the ground surface are more important in frozen periods.

Our study determined that the rainfall threshold is variable even in the same torrent due to the influence of the formation of frozen ground and changes in sediment storage volume. Therefore, the occurrence of frozen ground and the sediment storage volume need to be monitored and estimated to improve debris flow disaster mitigation in cold regions. Because periods of ground freezing and the sediment supply ratio are strongly affected by climatic conditions, future changes in the debris flow threshold should be carefully monitored for continuous sediment flow-associated disaster mitigation under climate change.

**Data availability**

Because this study was commissioned by Ministry of Land, Infrastructure, Transport and Tourism (MLIT), Japan, the ground temperature data are available from the corresponding author, Fumitoshi Imaizumi, upon reasonable request and agreement by MLIT. Rainfall and topographic data are available upon agreement by MLIT, which holds the copyright of the data.

**Author contributions**

FI and OO conceived and designed the study. AI designed and conducted field monitoring of ground freezing. KY contributed analyses of historical rainfall data. All of the authors conducted field motoring. FI contributed to the writing of the paper.

**Competing interests**

The authors declare that they have no conflict of interest

**Acknowledgements**

This study was funded by Ministry of Land, Infrastructure, Transport and Tourism, Japan. This study was also supported by
by JSPS Grant Numbers 18H02235 and 19K01156. Airborne DEMs and rainfall data was provided by Fuji Sabo Office, Chubu
Regional Bureau, Ministry of Land, Infrastructure, Transport and Tourism, Japan. We are grateful to Tomohiro Egusa for his
advice on statistical analysis and to Sohei Enomoto for his help to drill the hole for temperature monitoring. The author
acknowledges Francis Rengers and one anonymous reviewer for their constructive reviewers that improved this paper.

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
