# Peer review of "Temporal changes in the debris flow threshold under the effects of ground freezing and sediment storage on Mt. Fuji"

_Earth Surface Dynamics, 2021_

## Author Response (AR1)

Reviewer 1

We sincerely thank you for the efforts you have made to improve our submission to "*Earth Surface Dynamics*". We have responded to all review comments in the following paragraphs. The blue-highlighted sentences are the review comments; sentences in black represent our responses to these review comments. Labels and line numbers after our response correspond to those in the revised manuscript with tracked changes.

The manuscript (MS) deals with the role of (a) ground freezing and (b) sediment storage in debris-flow triggering and how rainfall thresholds are affected. Relatively long records (~50 years) of debris-flow observations in a Japanese gully are combined with ground-freezing periods estimated from a model, and sediment storage volumes estimated from LiDAR measurements or aerial photographs. While the role of sediment storage was less revealing, the authors found ground freezing to alter rainfall thresholds significantly and explain it with different infiltration rates depending on if the ground is frozen or unfrozen. The MS discusses an important topic with possible impacts both on the development of early warning systems and climate change impact assessments in cold debris-flow prone regions.

It is our great pleasure that the reviewer is interested in our study.

The MS is generally well written and structured. The data sets provide a good basis for the research on the influence of climate on debris-flow activity. The author's approach in estimating ground-freezing periods and connecting it to debris-flow triggering is original and promising. However, the analysis and discussion on other potential reasons for differences in rainfall thresholds are missing and therefore I had difficulties in believing the findings. The main reasons concern the following:

Seasonality in rainfall. I assume that the frozen periods mainly fall in spring and autumn. Are rainfall event characteristics also different in these seasons compared to summer? You could either estimate rainfall thresholds for your data split into seasons. They may look similar as the ones you already obtained and then you could argue that you obtain the same thresholds when splitting into frozen and unfrozen. Another way would be to show histograms of rainfall characteristics (e.g. mean intensity) for frozen and unfrozen periods. If they look similar, you know that the shift in rainfall threshold is not introduced by seasonal characteristics.

Rainfall characteristics are clearly different between frozen and unfrozen seasons. As shown in new Fig. 10, total rainfall and hourly rainfall intensity in frozen periods are higher than those in unfrozen periods. We have added new Fig. 6 to show seasonal changes in the rainfall intensity (lines 336-340; Rev.1-1). Fig. 2 also shows that rainfall depth in summer is significantly higher than that in winter. Because rainfall characteristics are different amongst seasons, it is difficult to conduct analysis

suggested by the reviewer.

At the same time, if seasonal rainfall pattern is the most important factor affecting timing of debris flow, debris flows occur more easily in unfrozen periods when precipitation is high (498-505; Rev.1-1). As described in this paper, however, the number of debris flow in frozen periods is larger than that in unfrozen periods (lines 341-349; Rev.1-1). In unfrozen periods, there are so many no flow events exceeding rainfall thresholds during frozen periods (new Figs. 9, 10). Therefore, rainfall threshold is affected by the formation of frozen ground rather than rainfall patterns. We have improved explanation on the seasonal rainfall pattern and difference in rainfall thresholds between frozen/unfrozen periods in order to clarify effect of ground freezing on occurrence of debris flow (lines 336-340, 498-505, 341-349; Rev. 1-1).

Antecedent rain/wetness and snowmelt. Another reason for lower ID thresholds could be the antecedent conditions. Antecedent rainfall and snowmelt are probably the most important control. Analysing antecedent rainfall can be done from your data. Are there snow depth measurements to constrain the melting season?

Unfortunately, there is no snow depth measurement around the initiation zone of debris flow. Clear diurnal changes in the ground surface temperature even in midwinter (Fig. 4) indicate that snow was not considered important to the formation of frozen ground at the monitoring point (line 352 in previous version of the manuscript). However, snow depth would be spatially variable affected by local topography. We have added statements about snowpack in the study site and result sections (lines 92-99, 186-192, 290-291,481-493; Rev.1-2). Based on the review comments, we have added a new analysis on the antecedent rainfall (new Figs. 6, 8). Antecedent rainfall is generally low in frozen periods when many debris flow occurs. Additionally, magnitude of antecedent rainfall is generally much lower than total rainfall triggering debris flow (Fig. 8). Therefore, antecedent rainfall is not very important factor in Osawa failure (lines 365-366, 370-375; Rev.1-2).

Uncertainties. They can be quite large, especially if based on only a few points. With CSI you have an objective measure to estimate ID-thresholds, which is good. However, when you calculate ID-thresholds with different data (frozen & unfrozen), the significance of these differences is unclear. Because some of these thresholds were estimated based on few triggering events, the thresholds can be very sensitive to individual data points. Ideally, you would assess uncertainties e.g. with bootstrapping. However, I realize that such an analysis is not useful for e.g. Fig 9c, where you have quite a few points. Hence, it would be useful to show uncertainties at least where there are e.g. >10 triggering events.

Thank you so much for your beneficial comment. We agree that uncertainties can be large, especially in the analysis with few debris flow events. We have evaluated uncertainties using bootstrapping (Figs.

9, 10, 11, 12; Rev.1-3). As a result of the analysis, the constant $\alpha$ in ID threshold (Eq. 4) was different between frozen and unfrozen periods, while slope β widely overlapped between the two periods (lines 137-139; Rev. 1-3). The frequency distributions of maximum hourly intensity and total rainfall thresholds obtained by bootstrapping did not overlap between the frozen and unfrozen periods (Figs. 910b, 910c), indicating that the difference in threshold was not affected by the presence or absence of specific rainfall events (lines 428-420; Rev.1-3). Uncertainties in the analysis related to the effect of volume of channel deposits were also assessed by bootstrapping (Lines 439-440, 445-446; Rev.1-3)

These points would need to be clarified in order to convincingly show the relevance of ground freezing on debris-flow triggering. Then I see it as a very valuable contribution to Esurf. Furthermore, there were some unclarities in the methods and the data sets used. Further comments related to this and other minor issues are listed below.

Specific comments
41: I think you mention wildfires as an example for the sensitivity of hydrological parameters (e.g. infiltration rate) but the connection is not clear when reading. If wildfires are not relevant for cold regions, which you focus on, I would focus on the hydrological properties of cold regions.
Wildfire itself is not important in cold regions. However, relationship between infiltration rate and occurrence of debris flows has been discussed mainly in wildfire studies. We have revised the sentence to emphasize relationship between infiltration rate and occurrence of debris flows (lines 43-45; Rev.1-4). We also add citations on the hydrological processes in cold regions (lines 46-52; Rev.1-4).

40-48: Is the hydraulic conductivity the main difference from frozen to unfrozen periods? What is the role of water storage capacity?
As the reviewer points out, decreases in the maximum water capacity by the formation of ice in sediment matrix is a potential factor affecting occurrence of debris flow. We have added statement on the water storage capacity (lines 50-52; Rev.1-5).

58: What process drives sediment production?
Rockfall, failure of basaltic lava, rockfall, dry ravel, soil creep are important sediment supply processes. We have clarified types of sediment supply process (lines 102-104; Rev.1-6).

86: 2737 mm y-1
We have revised based on the suggestion (line 90; Rev.1-7).

132: please specify, within 1 calendar day or within 24 hours?

One calendar day. We will specify (line 149; Rev.1-8).

137: Could you add a marker in the figure to show JMA station?

It is hard to mark JMA summit station, because the distance between the summit and the building of summit station is just about 5 m. We have improved statement about location of the JMA station (line 154; Rev.1-9). We also clarified location of summit station in the figure caption (Fig. 1)

141: TM (Fig. 1d)

We have added "Fig. 1d" after TM (line 158; Rev.1-10).

143: It is not ideal to match air with ground surface temperature. Could you please state how this affects your extrapolated temperature?

Amplitude of diurnal changes in the ground surface temperature is generally higher than that of air temperature. However, we used the empirical model using the air temperature, because it could nicely estimate ground surface temperature using air temperature. We have explained it in the text (lines 160-164; Rev.1-11).

143: Please explicitly state which model you are using and add some description regarding the main assumptions, required input, output, why was it developed or for what is it usually used? For which spatial and temporal scales is it usually used?

We used linear ordinary least squares regression. Input and output are the daily average temperature at JMA summit station and the daily average ground surface temperature at TM, respectively. The daily average ground surface temperature was used to estimate downward progress of the freezing and thawing fronts. We have revised explanation of the estimation method (lines 160-169; Rev.1-12). Spatial scale would highly depend on spatial non-uniformity of the ground surface temperature affected by local topography. It is difficult to specify initiation point of debris flows, because debris flows start at many points even in one survey period. (lines 155-159, Rev,1-12) Based on this comment, we have improved discussion on the spatial variability of ground temperature (lines 482-497; Rev.1-12).

149: Eq 1 & 2, please add the independent variable to the equation (e.g. T(t)) and units. Now it is difficult distinguish between variables and parameters. Do cf and cm vary with time? Although you cite the equations, it would be helpful if you would state the assumptions behind it. Is this the degree-day method you mention before? The meaning of cf and cm are not clear to me.

Eqs. 1 and 2 are the degree-day method. This is coming from an approximate solution of unsteady heat conduction equation. $c_f$ and $c_m$, which are derived from thermal conduction, moisture content

ratio, heat of fusion and other parameters, are constant. This method assumes that sediment characteristics and moisture content ratio are spatially and temporally constant. We have explained these assumptions (lines 164-178). We also add the independent variable as suggested by the reviewer (lines 170-171; Rev.1-13).

177: Caine (1980) was the first to use ID thresholds to my knowledge, so I would add the reference. You should also have "rainfall" somewhere in this sentence.

We have cited Canine (1980) in the sentence. We have added the term "rainfall" in the sentence (line 211; Rev.1-14).

183: How do you know if there was snow or not?

We do not have data on the absence or presence of snow cover. This sentence explains general determination of slush avalanche and debris flow. We have improved the sentence to clarify what we mean (lines 221-223; Rev.1-15).

187: I don't understand the reasoning for choosing 24 hours. First, I don't think the gully is much larger than other debris-flow prone gullies, or if so, please provide a comparison. Second, is it clear that water has to be concentrated from a large area from the headwaters, are debris flows triggered by runoff in this site? The chosen time period is not irrelevant because it affects the number of non-triggering rainfall events, thus also the CSI and thus the rainfall threshold.

Drainage area of the Osawa creek at the apex of alluvial fan is 11 km$^2$, larger than Chalk cliff (USA), Gadria (Italy), Kamikamihori, Ohya (Japan), Rebaixader (Spain) and other torrents. Although drainage area of Illgraben is similar to Osawa-creek as a whole, initiation zones of debris flow in Illgraben, which are dispersed in multiple sub-basins, are smaller than that is Osawa failure (about 1 km$^2$). Therefore, scale of hydrological processes in Osawa creek is likely larger than other debris-flow prone torrents. However, we do not have data which support adequacy of the 24 hours. We have used some other time period separating different rainfall events (e.g., 6 h ,12 h). As a result, CSI was highest when we set 24 h as a time period separating different rainfall event (lines 199-203, 362-366, 373-375; Rev.1-16, new Fig. 8).

194: 5 x 10 m or 5 x 5 m?

As we described, $10 \times 10$ m or $5 \times 10$ m is correct (line 241; Rev.1-17).

202: Looking at Figure 1 I have some troubles imagining the channel deposit dropping to zero. How is the justified? Is it really zero and the bedrock with outcropping bedrock or is it just the lowest value?

Geology in Osawa failure is alternation of basaltic lava and scoria. In the section with basaltic lava,

we can find exposure of bedrock in channel. However, it is sometimes difficult to distinguish channel deposit from scoria layer. Our analysis is based on the lowest elevation value. We have clarified that (line 251-253; Rev.1-18).

209: What are the expected consequences of these inherent errors?

This may affect misclassification of channel deposit class in the period with aerial photograph survey, especially in the period from 1993 to 1997 when volume of the channel deposits was just around the threshold of volume classes. We will note potential effects by these inherent errors (lines 263-265; Rev.1-19).

210: Please explain why you decided to determine two thresholds. Are there indications for that with <350'000 m3 supply is the limiting factor?

The boundary of the two classes was set to 350,000 $m^3$ to ensure that a statistically sufficient number of debris flow events could be allocated to both volume classes. Hence, this value does not have any physical meaning relevant to occurrence of debris flow. There is a possibility that rainfall threshold does not change exponentially at a specific volume of channel deposits. Therefore, it would be better to classify into many volume classes. However, it was not possible to conduct such analysis because insufficient number of debris-flow events, especially in unfrozen periods. This is limitation in our study. We have added explanation on this point in the text (lines 263-264; Rev.1-20).

215: Table 1, could you add the dates of the aerial photographs here? It is not easy to follow in the text when which data was obtained. I think the text does not refer to the table.

As suggested by the reviewer, we have added information of the aerial photograph in the table. If we add the dates of all aerial photographs, the table become very long. Additionally, based on the next comment, we have described timing of aerial photographs in Fig. 3. Therefore, we changed style of the table. We also referred Table 1 in the text (Table 1, Rev.1-21). We found that the type of topographic data prior to 1980 was not same as that after 1980. We have revised explanation in the text (line 240-242; Rev.1-21)

230: Figure 3. Could you add markers to dates with photograph/Lidar observations?

We had described timing of measurements in Fig. 3.

249: Could you please add markers for debris flows/slush avalanches?

Just two sediment transfer events were occurred in this period (March 5, 2018 and May 21, 2019). We have added markers (Fig. 4).

258: How did you calibrate? What measure did you optimize? I can't judge your model performance from this section, only visually from Fig. 5.

We selected $c_f$ that minimizes errors in estimation of downward progress of freezing line in the depths between 0.25 to 1.25 m. $c_m$ was calibrated based on the melting timing at a depth of 1.25 m. We have clarified calibration methods (lines 302-304, 314-315; Rev.1-22).

264: If it is a physical parameter, I guess it should have some unit, right?

Unit of $c_f$ and $c_m$ are m °C$^{-1/2}$ d$^{-1/2}$. We have added the unit (lines 304, 314; Rev.1-23).

268 – L. 280: Unfortunately, I don't see how this section fits into the story. Why do you only show one example of channel change? What happened to the other Lidars? How can you generalize from this figure that all debris flows sourced from channel deposits?

Total 68 debris flow initiation points were detected by the analysis of airborne LiDAR data in the period from 2008 to 2017. Hence, a large space for topographical and statistical analyses is needed to generalize findings in this section. Although it is worthful to deepen topographic analysis, we afraid that aim of this paper will be obscured by adding such analyses. Furthermore, as the reviewer comments, this section is not closely related to the main topic of the paper. Therefore, we have removed this section from revised manuscript (line 321; Rev.1-24).

283-387: These two result sections are very clear (with reservations mentioned in the general comments).

Thank you so much for your comment.

349: If you say what was not considered, please also mention what your model considers.

Our model considers ground surface temperature. We have revised the sentence (lines, 487; Rev.1-25).

354: This is true for the ridge, but aren't wind and incoming solar radiation conditions different in the creek? Can you say something about the distribution of snow within the creek?

As the reviewer points out, snow depth at the bottom of gully is deeper than that at ridge and side slopes of the gully. Unfortunately, we do not have data on the spatial distribution of snow depth in the creek. We have added a discussion on the distribution of snow depth (lines 482-494; Rev.1-26). Although ground freezing and snow conditions at very bottom of the valley may differ from TM, effect of ground freezing could be demonstrated by the estimation of freezing periods at TM (Figs. 9, 10). This implies that water supply condition on hillslopes, which have greater area than valley bottom, is important to occurrence of sediment transfer. We have added discussion on this point (lines 494-497; Rev.1-26)

356: What is your definition of "significantly different"?

We have deleted the description in relation to the other review comments (Rev.1-27). We hope difference in the rainfall threshold are statistically revealed by bootstrapping.

377: Figure 10. In the light blue frozen layer, are the pores filled with ice? Nice schematic!

Light blue layer indicates frozen ground. We think volumetric ratio of ice in the soil is different amongst depths and seasons. On March 5, 2018, infiltration of rainfall water rapidly increased ground temperature in the deep layers (1.5 m), implying that a part of pores was not filled with ice. We have explained that ice exist soil matric in the lite blue layer (Fig. 10 caption, Rev.1-28).

394: why is this ratio controlling the movement?

It is because the ratio controls balance between shear stress and shear strength of cohesionless sediment (Imaizumi et al., 2016; 2017). We have improved the sentence (line 547-549; Rev.1-29).

431: I think a detailed explanation is required for why the underlying data is not publicly available

We have added explanation in "Data availability". Because this study was commissioned by Ministry of Land, Infrastructure, Transport and Tourism (MLIT), Japan, the ground temperature data that support the findings of this study are available from the corresponding author, Fumitoshi Imaizumi, upon reasonable request and agreement by MLIT. Rainfall and topographic data are available upon agreement by MLIT, which holds the copyright of the data. (594-597; Rev.1-30).

Figures 1 & 6 are missing coordinates

We have added coordinates in Fig. 1. We have deleted Fig. 6 based on another review comment.

Reviewer 2 (Dr. Francis Rengers)

We sincerely appreciate the efforts you have made to improve our submission to *Earth Surface Dynamics*. Comments from the reviewer are very helpful for us to improve our manuscript. We have responded to all review comments in the following paragraphs. The blue-highlighted sentences are the review comments; sentences in black represent our responses to these review comments. Labels after our response correspond to those in the revised manuscript with tracked changes.

This is a very well written manuscript with useful figures, clear language, and good supporting data. I think the researchers have done a nice job explaining a relatively under-studied phenomena that will be of general interest. I have a few general suggestions that I think might help to increase the impact of this paper.

It is our pleasure that the reviewer is interested in our study. We have improved our manuscript based on review comments.

First, for your debris flow thresholds, many people use either a lower limit or an upper limit to separate debris flows from non-debris flows (Staley et al., 2017; Tang et al., 2019). Consider something like for figures 7-9. In addition, consider putting using a dimensionless discharge criteria to see if that helps to separate out your slush avalanches from your debris flows (e.g. Tang et al., 2019).

Thank you for your beneficial comment. As the reviewer points out, many researchers use a lower limit or an upper limit threshold. We have added lower threshold of the debris flow occurrence in new Fig. 9. Some plots of no flow event distribute upper-right side of the sediment transfer plots in the ID diagram, especially in unfrozen periods. Therefore, upper limit of rainfall events that does not produce sediment transfer could not be determined in the Osawa failure (lines 389-384; Rev.2-1). We could not obtain upper and lower limit line in the analysis considering volume of channel deposits because of the limited number of debris flow samples. Although discharge has been monitored at some observatories along the Osawa Creek, they are far from initiation zone of debris flow. Furthermore, there is no discharge at observatories during rainfall events without debris flow. We think it is difficult to clarify initiation process of debris flow in Osawa failure by the method like Tang et al., 2019, GRL.

Second, is that you might want to discuss the influence of aspect on your results (see Tillery and Rengers, 2019 for some general considerations of debris flow thresholds and aspect). You could also use an approach similar to Hales and Roering to estimate/model the temperature effects of aspect, but that might be too much work at this point.

Total 68 debris flow initiation points were detected by the analysis of airborne LiDAR data in the period from 2008 to 2017. Slope aspect of almost all of debris flow initiation points was west (more

properly north west to south west) because of the orientation of the Osawa failure (please see contour lines in Fig. 1d). Therefore, it is hard to show detailed analysis on the slope aspect. However, we think slope aspect is an important factor controlling occurrence of debris flow. We have added discussion on the slope aspect in the revised manuscript (lines 558-561; Rev.2-2).

Third, can you address the fact that it probably rains the most when there is less frozen sediment because in the winter it is mostly snow. In addition, you should probably acknowledge that when it rains it will melt some of the ice. If the amount of melt can be estimated, it will help to strengthen your argument.

Based on the suggestion, we have described that it is snow in mid-winter (341-344; Rev.2-3). In order to clarify that snowfall does not directly trigger sediment transfers (debris flow and slush avalanche), we have added a new figure showing monthly frequency of sediment transfer (new Fig. 7). It is very difficult to evaluate amount of ice melt. However, we have added a discussion that water supply by ice and snow melt may affects occurrence of debris flow and slush avalanche (lines 512-513, 521-523; Rev.2-3).

Fourth, consider making some plots like Figure 4 (b and c) from Staley et al., 2017.   I'm wondering if you might see more separation if you focus on a single rainfall intensity.

Thank you for your suggestion. We have added a figure showing time series of rainfall intensity like Staley et al., 2017, Geomorphology (new Fig. 6). Debris flow and slush avalanche events are not simply separated from no flow events by the rainfall intensity. Therefore, other factors, such as total rainfall, rainfall duration, ground freezing, volume of channel deposits, affect debris flow occurrence together with rainfall intensity. We have described the difficulty in separating debris flow events only by rainfall intensity (lines 344-348; Rev.2-4)

Finally, I think it'd be nice if you could add some language to the discussion to talk about how warming would effect debris flow generation, this would help to show how your work would be more globally impactful in high alpine areas.

Previous studies have implied impact of climate change on debris flow activity in cold regions via changes sediment supply rate and increasing frequency of extreme rainfall events (Staffler et al., 2008; Stoffel et al., 2014a, 2014b). Global warming affects seasons with frozen ground, possibly changing seasonal debris flow activity in cold regions. We have discussed at the end of discussion (lines 569-572; Rev.2-5).

Specific Comments:

Line 51: I would add this recent reference that discusses debris flow material.
Rengers, F.K., Kean, J.W., Reitman, N.G., Smith, J.B., Coe, J. A., McGuire, L.A. 2020. The Influence of Frost Weathering on Debris Flow Sediment Supply in an Alpine Basin. Journal of Geophysical Research: Earth Surface. https://doi.org/10.1029/2019JF005369

We have cited the recommended paper that discuss debris flow material (line 55; Rev.2-6).

76: What do you mean by size (depth? Length?)

We mean length and width. We have clarified (line 80; Rev.2-7).

86: When you say >100mm what time duration are you talking about: hour, week, month, year? Seasonally? For a single storm?

Total rainfall in a single storm. We have clarified (line 90; Rev.2-8).

116: Here are you talking about a peak discharge? Or the total discharge over time? If you include units that would help to clarify.

This is total discharge in a single event. The unit is $m^3$. We have improved the description (lines 134-135; Rev.2-9).

141: Define the degree-day method

The degree-day method is expressed as Eqs. (1) and (2). We have improved explanation of the degree-day method (lines 164-166; Rev.2-10).

238: You might want to explain why this happens somewhere. The temperature should decrease from the surface as you move deeper up to a certain point, but then the temperature will slightly increase with depth due to radioactivity.

Our sentence or Fig. 5 might have confused the reviewer. Although the freezing front goes down from the ground surface, the daily average temperature at the surface was generally lower than the deeper part during winter (Fig. 4). This depth-ward increase in temperature restricts the maximum depth of the frozen layer to 1.5 m at the monitoring site.

The reviewer might have got interested in the temporal pattern that the 1.5 m-deep point thawed much earlier than the 1.25 m-deep point in Fig. 5. The depth of 1.5 m was frozen but was never lower than the melting point (Fig. 4), which means the water phase at the depth was quite sensible to a slight difference between heat output toward the surface (potentially decreasing temperature) and geothermal heat flow (potentially increasing temperature). In the observed case, geothermal heat thawed ground upward around the deepest position of the seasonal frozen layer. However, the most part of the frozen layer thawed downward because the heat input from the surface is much larger than the geothermal

heat flow. We will add some sentences explaining Figs 4 and 5 in our revised manuscript (lines 280-282, 316-318; Rev.2-11).

271: Make sure to say if this is a topographic difference from lidar or something else.

Because errors in the LiDAR survey (< 1m) is less than the difference of DEMs (> 5 m), changes in the topography due to sediment supply occurred in this area. However, we removed this section based on a comment by the other reviewer.

290: One question people might ask is why rain doesn't melt ice that is between sediment particles, so I think it'd be good if you could address that.

A previous study written in Japanese revealed that some heavy rainfall events during summer promote thawing of frozen layer on the summit of Mt. Fuji, although such thermal disturbance was unclear for many rain events. We add a sentence of thawing potential by rain in the revised manuscript (lines 310-313; Rev.2-12).

296: Where you say "maximum rainfall intensity", what duration are you using (1 hour, 30 min., etc.?)

This is rainfall intensity in 1 hour. We have clarified (lines 367-368; Rev.2-13).

389: When you say channel deposits, do you mean the sediment stored in the channel or the sediment that is removed from the channel and deposited downstream?

We mean sediment stored in channel. We will clarify (line 341-342; Rev.2-14).

---

## Author Response (AR2)

**Reviewer 1**

We sincerely appreciate your efforts to improve our submission to "*Earth Surface Dynamics*". We have responded to all review comments in the following paragraphs. The blue-highlighted sentences are the review comments; sentences in black represent our responses to these review comments. Labels and line numbers after our response correspond to those in the revised manuscript with tracked changes.

The manuscript has substantially improved and I was glad to see the added analyses on rainfall seasonality and uncertainty. I look forward to seeing this study published and think that it will draw some attention to this underrepresented topic of runoff formation in cold regions and it's relation to hazards. However, before publishing in Esurf I have one general comment and more specific ones are listed further down.

Thank you for your comment. We think our manuscripts has been improved by your review comments.

The authors repeatedly point to the importance of their findings with regard to climate change (e.g. L.39., L522, L. 541). However, I am missing a discussion on possible implications of the findings for the future debris flow hazard. I realize that is complex, as it depends on changes in temperature, precipitation, sediment recharge, etc. and not all processes will consistently change towards more or less debris flows. Nevertheless, it is important because as it is now, one could conclude that a warmer climate increases debris flow rainfall thresholds (Fig. 10) and additional sediments due to e.g. permafrost thawing would stabilize the debris flow material (Fig. 12).

Based on the review comment, we have added discussion on the impact of future climate change on debris flow hazards (Rev.1-1, lines 539-545). Climate change has both positive and negative effects on debris flow hazards. Our study implied that debris flow risk decreases by the climate change by shortening the periods with seasonal frozen ground. We have added a statement on climate change in conclusion (Rev.1-1, lines 555-556).

Specific comments

L.20: I would consider leaving out the sentence in the abstract on higher thresholds with more sediment storage. First, it is counter-intuitive and would need an explanation. Second, it is based on quite few events.

We have removed the sentence based on the suggestion by the reviewer (Rev.1-2, line 20).

L.27: Consider citing Hirschberg et al. (2021) https://doi.org/10.1029/2020JF005739

We have cited Hirschberg et al. (2021) as suggested by the reviewer (Rev.1-3, line 28). We also cited this paper in discussion (Rev.1-3, line 544)

L.67-73: This paragraph jumps from aims to methods and then to aims again and may not be clear to the reader. I would start with the goals and then explain how you get there. For example, the focus is on "...understand the hydrological processes triggering debris flows..." so I would put this first and then say something like "We do this by analyzing temporal changes in rainfall thresholds. To this end we estimated seasonal ground freezing ground freezing and use it for...".

We have revised structure of the last paragraph in Introduction. We started with aim of the study and then explained methods used to achieve the aim (Rev.1-4; lines 67-77).

L.125: Which "problems" do you mean?

The problems include electric troubles and destruction of devices. We have described in the sentence (Rev.1-5, lines 129-130).

L.167-169: it would be helpful if you could provide units for all variables and parameters. If possible, also typical values for cf and cm.

We have clarified units of variables and parameters (Rev.1-6, lines 172-176). The value cf generally ranges from 0.02 to 0.06. However, it is difficult to show typical value for cm. Therefore, we did not describe typical values.

L.213: How did you bootstrap? did you resample to have the same number of triggering and non-triggering events as the original observations?

Total number of resamplings was set to the same number of rainfall events as the original observations under each condition of debris flow initiation zone. Number of debris flow events is not consistent among bootstrapings. We have clarified it in the text (Rev.1-7, lines 119-224).

L.221: please explain why this decay factor is necessary

We used decay factor (K<1), because a considerable amount of groundwater gradually infiltrates into deeper part of the mountain body in high elevation on Mt. Fuji. We have explained in text (Rev.1-8, lines 235-237).

L.235: I'm not sure how to interpret this. Does that mean there is a bias? I think providing the standard error would correspond better to other studies.

This is standard errors between control points along horizontal and vertical directions. We have

clarified that these are standard errors (Rev.1-9, line 247). There was a bias between horizontal and vertical directions in some periods. Because we could not access original data, it was not possible to calculate the simple standard error.

As suggested by the reviewer, we have made the major scales longer (Fig. 3).

As the reviewer points out, maximum value is $160*10^4$ m$^3$. We have revised the figure (Fig. 3).

GS indicates ground surface. We have revised the figure legend (Fig. 4).

According to calculation formulas, the cf, which minimizes standard error, also gives minimum mean square error and root mean square error. Therefore, we can also say root mean square error instead of standard error. For the time being, we did not revise the term.

We have added a statement that the antecedent rainfall was determined using Eq. 5 (Rev.1-10, lines 339-340). We have changed direction of sales on x-axis (Fig. 6).

We have deleted "in the figure" (Rev.1-11, line 339).

We have added "s" after the "threshold" (Rev.1-12, line 350).

We have improved explanation of the bootstrapping (Rev.1-13, lines 219-224). We obtained appropriate rainfall thresholds in each resamplings.

L. 503: This is a good hypothesis for your findings. However, to avoid any misunderstandings I would also say that in case of threshold exceedance, which happens sooner or later, debris flow volumes will also be larger. As it reads now one could infer that more loose sediments mitigate debris flows. In addition, this specific finding could also result from the fact that there are relatively few triggering events.

Based on the suggestion by the reviewer, we have described that the debris flow volumes are likely larger when a large volume of channel deposits is stored in the channel. We also discussed possibility that relatively small number of debris flow events affected the specific characteristics of rainfall threshold in unfrozen periods (Rev.1-14, lines 517-520).

L.521: Please then describe how future seasonal ground freezing will affect debris flows.

We have added discussion on impact of climate change on occurrence of debris flow (Rev.1-15, lines 537-545). Because rainfall thresholds in unfrozen periods were higher than that in frozen periods (Figs. 9, 10), global warming possibly decreases debris flow risks by shortening the period with seasonal frozen ground, which decreases rainfall threshold for the occurrence of debris flow. Increases in the amount sediment delivered to channels by thawing of permafrost possibly increases volume of sediment storage in debris flow initiation zone, resulting in the higher rainfall threshold for the occurrence of debris flow in unfrozen periods. However, climate change also has effects that increases debris flow risks.

L.531-536: Again, the way it is written one could think that e.g. increased amounts of sediment form instable slopes due to permafrost thaw would increase the threshold and mitigate debris flows. But climate change will also provide loose sediments where there was none before.

As described above, we have added the discussion on the climate change (Rev.1-15, lines 537-545). In addition, we have added a sentence on the climate change in conclusion (Rev.1-16, lines 555-556)
.